



# Observation-driven model for calculating water harvesting potential from advective fog in (semi-)arid coastal regions

Felipe Lobos-Roco[1,2], Jordi Vilà-Guerau de Arellano[3], and Camilo del Río[4]

[1]Centro UC Desierto de Atacama, Pontificia Universidad Católica de Chile.
[2]Facultad de Agronomía y Sistemas Naturales, Pontificia Universidad Católica de Chile.
[3]Meteorology and Air Quality group, Wageningen University, the Netherlands.
[4]Instituto de Geografía, Pontificia Universidad Católica de Chile.

**Correspondence:** Felipe Lobos-Roco (flobosr@uc.cl; felipe.lobos.roco@gmail.com)

**Abstract.**

Motivated by finding complementary water sources in (semi-)arid regions, we develop and assess an observational-driven model to calculate fog harvesting water potential. We aim to integrate this model with routine meteorological data collected under complex meteorological and topographic conditions to characterize the advective fog phenomenon. Based on the mass

balance principle, the **A**dvective fog **M**odel for (semi-)**A**rid **R**egions **U**nder climate change (AMARU) offers insights into fog water harvesting volumes in time and space domains. The model is based on a simple thermodynamic approach to calculate the dependence of the liquid water content ($q_l$) on height. Based on climatological fog collection records, we introduce an empirical efficiency coefficient. When combined with $q_l$, this coefficient facilitates the estimation of fog harvesting volumes (L m$^{-2}$). AMARU's outputs are validated against in-situ observations collected over Chile's coastal (semi-)arid regions at various

elevations and years (2018-2023). The model's representations of the seasonal cycle of fog harvesting follow observations with errors of $\sim$10%. The model satisfactorily estimates the maximum $q_l$ ($\sim$0.8 g kg$^{-1}$) available for fog harvesting potential in the vertical column. To assess spatial variability, we combine the model with satellite-retrieved data, enabling the mapping of fog harvesting potential along the Atacama coast. Our approach enables the application of the combined observational-AMARU model to other (semi-)arid regions worldwide that share similar conditions. Through the quantification of fog harvesting, our

model contributes to water planning, ecosystem delimitation efforts, and the study of the climatological evolution of cloud water, among others.

## 1 Introduction

Water resources in (semi-)arid regions are of critical value for social, economic, and ecological development. However, in recent decades, climate change has enhanced drought periods, intensifying water stress in areas already facing scarcity. This

has resulted in a worldwide dryland expansion (Koppa et al., 2023). For example, Chile's (semi-)arid and mediterranean regions have suffered a fifteen-year drought, experiencing a nearly 40% decrease in precipitation (Garreaud et al., 2021). Likewise, other dry regions such as California, South Africa, Australia, Spain, and Morocco are confronting similar challenges related to water scarcity, including new threats like fire risks, degradation of soil ecosystems, and impacts on food security (Goulden and





Bales, 2019; Berbel and Esteban, 2019; Keeley and Syphard, 2021; Kogan and Kogan, 2019). Moreover, future IPCC's climate

scenarios are discouraging, projecting even drier conditions by 2050 (Masson-Delmotte et al., 2021). Under this escalating water scarcity scenario, the exploration of new water resources is imperative. In this context, the collection of freshwater from fog presents itself as a viable alternative to face water scarcity, especially in (semi-)arid regions along the subtropical western coasts. Fog harvesting has long represented a significant untapped water potential in the world's dry regions (Klemm et al., 2012). For example, in the coastal Atacama Desert, fog and dew represent the sole water source across vast territories with

almost null precipitations (Cereceda et al., 2008). However, quantifying this water potential represents a scientific challenge, requiring a deep understanding of the physical processes controlling the formation and dissipation of the marine stratocumulus (Sc) cloud deck over the ocean (Andersen et al., 2020; del Río et al., 2021b), its interaction with coastal topography (Lobos-Roco et al., 2018), and the effectiveness of fog collector designs (Verbrugghe and Khan, 2023). In addition, the lack of available and direct observations of the fog phenomenon, combined with the complexity of topography, makes it challenging to pinpoint

where fog forms, identify optimal harvesting seasons, and determine potential yield. Consequently, advancing our knowledge to quantify harvestable water from fog clouds is imperative to develop this promising alternative water source. Estimating where, when, and how much water can be harvested from fog is socially relevant. Estimating fog water potential can facilitate the transition from experimental fog harvesting practices to industrial ones (Lobos-Roco et al., 2024), potentially enhancing the development of overlooked desert territories, and benfiting their local communities. Moreover, estimating potential fog water

production can help us better understand the unique ecosystems sustained by fog (Koch et al., 2019; Muñoz-Schick et al., 2001; Moat et al., 2021), contributing to the assessment of their conservation status under a rapidly warming climate. Fog is a meteorological phenomenon defined by a boundary layer cloud in permanent contact with the Earth's surface (Roach, 1995; Stull, 2012). The origins of fog are, influenced by different atmospheric boundary layer and local topographic conditions. However, in most of the (semi-)arid regions along the (sub)tropical western margins of continents, fog formation is driven

by the advection of the Sc cloud. The Sc cloud forms over the ocean in a vast deck influenced by sea surface temperature and large-scale subsidence (Wood, 2012), whose interaction results in a strong thermal inversion layer (Muñoz et al., 2011). The stability of the marine boundary layer (MBL) determines the formation, maintenance, and dissipation of the Sc cloud. Formation and maintenance depend on how well-mixed ($<3.1$ x $10^{-3}$ K m$^{-1}$) the MBL is, while the dissipation is influenced by its stratification ($>3.1$ x $10^{-3}$ K m$^{-1}$) (Lobos-Roco et al., 2018). This cloud forms at the upper part of the MBL, exhibiting

a clear vertical structure. This structure is characterized by an averaged cloud base ranging from 300 to 400 m (Lu et al., 2007), determined by the lifting condensation level. As the latter increases, the liquid water content progressively rises, peaking at the cloud top ($\sim$0.7 g kg$^{-1}$) (Schween et al., 2022). The liquid water content abruptly drops to 0 g kg$^{-1}$ just above the cloud top, where the air becomes stratified and extremely dry. The Sc cloud is advected into the continent by the typical strong thermal-driven sea breeze of (semi-)arid regions (Lobos-Roco et al., 2021). Upon reaching land, the cloud deck is affected by

local conditions that, together with the high topography, lift it, forming fog belts (del Río et al., 2021b). Depending on latitude and topography, these fog belts vary in altitude; for example, in the Atacama region, they are found in the coastal mountains between 600 to 1200 m ASL (Cereceda et al., 2008; Garreaud et al., 2008). This narrow belt represents the area where fog can potentially be harvested. This harvesting process is already made by nature through specialized plants that accumulate water in





their leaves, spines, and branches, making it available for the soil and roots (Malik et al., 2014; García et al., 2021; Koch et al., 2019). Likewise, artificial collectors have been developed to efficiently harvest fog water using meshes (Schemenauer and Cereceda, 1994). Numerous studies have reported promising fog harvesting volumes worldwide in arid and semi-arid regions. For example, rates between 6-8 L m$^{-2}$ d$^{-1}$ have been reported in the hyperarid Atacama Desert in Chile (Cereceda et al., 2002; Larrain et al., 2002), 1-5 L m$^{-2}$ d$^{-1}$ along the western coast of South Africa (Klemm et al., 2012), and 7 L m$^{-2}$ d$^{-1}$ in the Iberian Peninsula, Spain (Estrela et al., 2009).

In recent years, significant progress has been achieved in understanding the spatial variability of Sc cloud and fog (del Río et al., 2021b; Andersen et al., 2020), as well as the vertical structure of the fog cloud (García et al., 2021; Lobos-Roco et al., 2018) and the practical applications of fog and dew collection in water-stressed regions (Lobos-Roco et al., 2024; Baguskas et al., 2021). Despite these advancements, there remains a need to integrate these findings into a unified model that can address the questions of where, when, and how much water can be harvested from clouds. In this research, we present the **A**dvective fog **M**odel for (semi-)**A**rid **R**egions **U**nder climate change (AMARU), a phenomenological model designed to estimate fog harvesting potential volumes continuously in time and space.

## 2   Model formulation and evaluation

The AMARU aims to estimate the liquid water capacity of Sc clouds and the potential for fog harvesting using widely available routine meteorological data. Based on phenomenological principles, Figure 1 schematizes the physical interpretation, including all variables, dimensions, and processes. The model relies on the mass conservation principle in dimensional analysis. When fog occurs, a certain amount of water is retained from the total specific humidity ($q$) that passes through a passive collector. This implies that the harvested water results from the difference between water input and water output. This equation reads as follows:

$$\frac{dq_h}{dt} \simeq \frac{\Delta q_h}{\Delta t} = q_{in} - q_{out}, \tag{1}$$

here, $q_h$ is the fog harvesting capacity, $q_{in}$ is the total humidity available in the air for harvesting by the collector, and $q_{out}$ is the amount of humidity that remains unharvested. We introduce three assumptions: (1) the total specific humidity corresponds to the sum of saturated specific humidity and air-liquid water content, expressed as $q = q_s + q_l$; (2) given that $q_l$ is by two orders of magnitude lower than $q_s$, we approximated $q_{s(in)} \simeq q_{s(out)}$; (3) $q_{in} > q_{out} > q_h$.

With these assumptions, we define the water input as the amount of liquid water present in the air ($q_l$), while $q_h$ is a fraction of this total water input. To combine the model with routine observations, we express Equation (1) in net terms as:

$$\frac{dq_h}{dt} = u q_l A \eta, \tag{2}$$



Where $u$ is the perpendicular wind speed (m s$^{-1}$) relative to the collector, facilitating the transport of $q_l$ (g kg$^{-1}$) into an area ($A=\partial y\partial z$, m$^2$) determined by the collector. The collector's permeable surface has a collection efficiency coefficient ($\eta$). For operationalization, the units of $q_h$ are expressed in L m$^{-2}$ (equivalent to mm) over a specific time period, once transformed by the air density (kg m$^{-3}$). Further details regarding the derivation from equations (1) to (2) can be found in Appendix A.

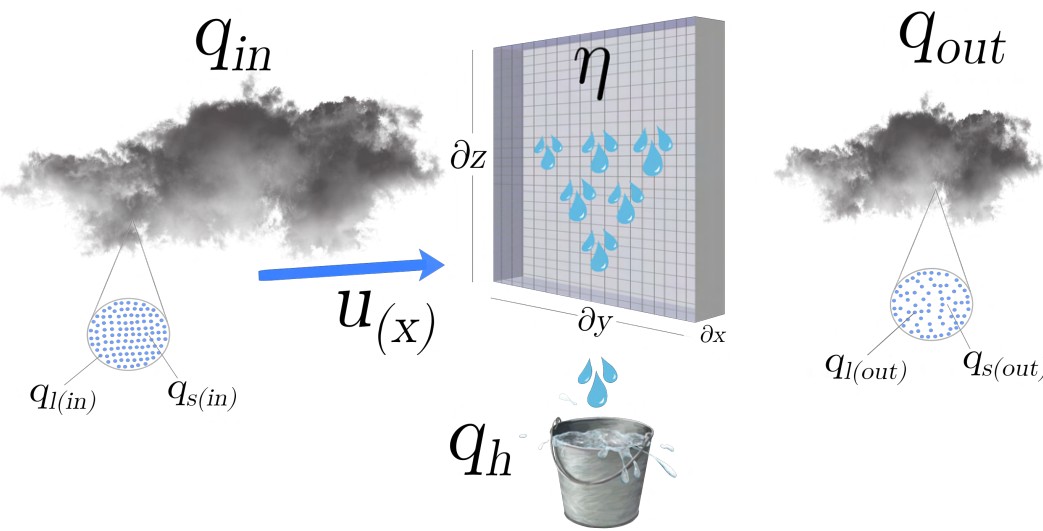

**Figure 1.** Physical interpretation of the equations (1) and (2).

In Equation (2), two terms are particularly sensitive: $q_l$ and $\eta$. The former varies in height (vertical dimension of the model) and depends on the conditions of the marine Sc cloud over the ocean and its interaction with the topography. The second term, $\eta$ is related to cloud microphysics, as well as the design and material properties of the collector. To delve into the details of $q_l$ and $\eta$, we deconstruct Equation (2) into two parts: the thermodynamic and water potential modules. Additionally, to study the model's horizontal spatial variability, we introduce a third module in this section, explaining the spatial interpolation of $q_h$ to create a fog harvesting potential map.





## 2.1 Thermodynamic module: obtaining liquid water content ($q_l$)

Liquid water content is a complex variable to estimate and measure. It can be obtained from complex and computationally expensive atmospheric models (Large-Eddy Simulation (LES); Weather Research and Forecasting (WRF)) (Bergot, 2016) or by
sophisticated and expensive instrumentation (fog monitors; microwave radiometers) (Kim et al., 2022). However, our objective here is to estimate $q_l$ using routine meteorological data. To achieve this, we propose employing the air parcel method (Wetzel, 1990), which calculates thermodynamic changes related to an air parcel while it is uplifted from the surface. This method has been successfully tested in the Atacama region by Lobos-Roco et al. (2024, 2018), who utilized two meteorological stations located at different heights ($z_1$ and $z_2$) along a topographic transect. This strategy allows for observation at two points within
the MBL during advective fog events: $z_1$ represents near-surface marine conditions, while $z_2$ represents inland conditions close to the MBL top, where fog formation occurs. Figure 2a schematizes the strategy for estimating $q_l$ using the parcel method. This estimation involves four steps, which are described and evaluated in the following subsections.

### 2.1.1 Fog cover fraction frequency

AMARU is a phenomenological model that relies on the presence of advective fog, which typically occurs under a well-mixed
MBL regime (Lobos-Roco et al., 2018). Thus, we propose three criteria for fog estimation using routine meteorological data. The first criterion posits fog occurrence when air temperatures reach the dew temperature ($T_a$-$T_d = 0$ ). However, this condition has been rarely observed, particularly in the coastal Atacama, even during fog formation. For this reason, we propose and test four alternative thresholds. For this estimation, we exclusively use data from station $z_2$. The second criterion is that MBL must be well-mixed. Our criteria for fulfilling this assumption is that the thermal gradient ($\partial\theta/\partial z$) between $\theta_{(z1)}$ and $\theta_{(z2)}$
is minimal. Here, we propose and test four thresholds close to 0 K m$^{-1}$. The third criterion is similar to the second one but employs the specific humidity vertical gradient ($\partial q/\partial z$) to assess MBL mixing. Similar to the first criterion we propose and test four thresholds to determine how well-mixed is the MBL in terms of temperature and specific humidity.

Figure 3 shows a statistical comparison between the estimated fog occurrence (in %) derived from the three proposed criteria and thresholds, and observations obtained from a Standard Fog Collector, SFC (Schemenauer and Cereceda, 1994).
The observations were conducted in the fog oases of Alto Patache ($z_2$) within the Atacama Desert during the year 2018 (20.82° S; 70.14° W; 850 m ASL, 5 km from the coast). In addition, we also use data from the meteorological station at Diego Aracena Airport, $z_1$ ( 20.52° S; 70.15° W; 48 m ASL), to calculate the vertical gradients.

In general terms, among the three proposed criteria, those based on $T_a$ -$T_d$ (blues) show the strongest correlation with directly observed fog collection. Among these, the threshold $T_a$ -$T_d < 1.5$ k (n.4 in Fig. 3a) emerges as the most accurate,
exhibiting a standard deviation aligned with observations (18%), a correlation coefficient (r) of 0.95, and a root mean square error (RMSE) of 6%. However, the remaining thresholds yield similar results, suggesting that fog occurs when $T_a$ -$T_d$ spans from 2 to 1.15 K. The second and third criteria are based on the parcel method and the mixing layer theory, which states that Sc cloud formation occurs under well-mixed MBL conditions. The chosen thresholds have been studied before in the coastal Atacama region by Lobos-Roco et al. (2018, 2024), del Río et al. (2021a) and García et al. (2021). The second criteria (depicted

**Figure 2.** (a) Schematic vertical cross-section representing the estimation of liquid water content ($q_l$) using the air parcel method. (b) Physical representation of the topographic uplifting of Sc cloud and its interaction in the ocean-land transition. Blue (orange) arrows indicate latent heat flux (sensible heat flux) from the surface (c). Representation of the combined meteorological conditions from stations $z_1$ and $z_2$ in the cloud base, cloud top, and $q_l$ representation.



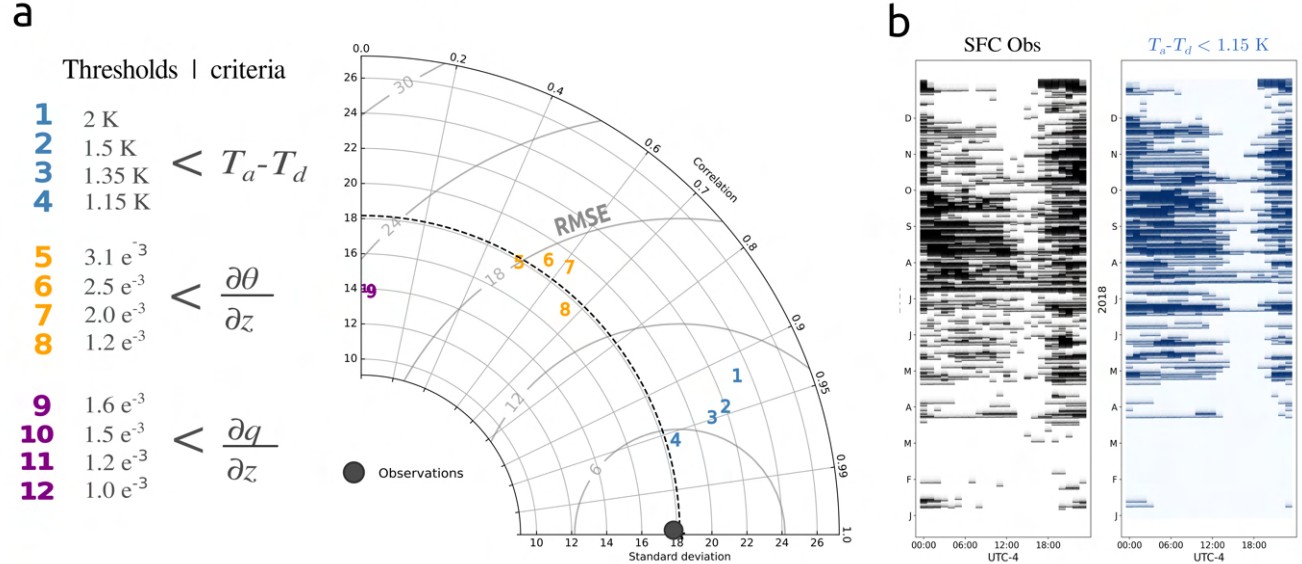

**Figure 3.** (a) Taylor diagram comparing the proposed criteria and thresholds for estimating fog occurrence. The diagram deploys correlation (r), standard deviation, and root mean square error (RMSE) between the criteria-thresholds and observations. (b) Comparison of the annual diurnal cycle of fog occurrence between observations (SFC) and the best-performing criteria.

in orange) show promising results when compared to observations, displaying a standard deviation ranging between 17 to 20%, an $r$ value ranging from 0.5 to 0.7, and a RMSE of ∼17%. These values suggest that the MBL tends to be thermally well-mixed (exhibiting minimal vertical gradients) during fog occurrences. The last criteria (purple ones) demonstrate insufficient performance in detecting fog occurrence, exhibiting no correlation with observations. This suggests that fog occurrence is not contingent upon MBL being well-mixed in terms of moisture. The disparity in the correlation between thermal and moisture vertical gradients with fog frequency can be attributed to the aridity of the observation location. On one hand, the arid terrain thermally contributes less to the MBL during fog events (low radiation), showing well-mixed MBL. Conversely, when fog is absent, the arid terrain contributes more thermally, leading to a stratified MBL. On the other hand, the arid landscape does not contribute moisture to the MBL during fog occurrences, nor does it when fog is absent, thereby showing no correlation with fog frequency. Figure 3b illustrates the diurnal cycle of fog frequency observed at the Alto Patache fog oasis throughout the year 2018, as measured by the Standard Fog Collector (SFC) and estimated using the threshold with the best performance (number 4). This threshold successfully estimates fog occurrence using simple meteorological data for any day and time throughout the year.

### 2.1.2 Cloud Base, CB

Once fog occurrence is estimated, we proceed to calculate the height of the fog-cloud base (CB). This process is summarized in Figure 2. The calculation assumes that the lifting condensation level (LCL) in boundary layer clouds such as Sc is equivalent to



the cloud base. To compute this, we adopt two approaches inspired by the parcel method of (Wetzel, 1990). The first approach solely considers data from the lowest station ($z_1$), representative of surface-marine conditions, where LCL corresponds to the height at which $q-q_s$=0 (Fig. 2a). This LCL represents the CB over the ocean.

The second approach considers two physical processes involved in the Sc-to-fog transition: environmental mixing and topo-
graphic uplifting. To represent the mixing with the environment experienced by an air parcel during adiabatic ascent, and based on (Lobos-Roco et al., 2018), we combined the meteorological conditions measured at both transect stations ($z_1$, $z_2$) using a mixing parameter $m$ as follows:

$$\psi_{p(z)} = (1 - m\frac{z}{z_{LCL}})\psi_1 + (1 - m\frac{z}{z_{LCL}})\psi_{1,2}, \tag{3}$$

Where $\psi$ is a scalar for potential temperature ($\theta$) or specific humidity ($q$), sub-fix $p$ represents the air parcel, numbering
sub-fix indicates the station used ($z_1$, $z_2$), $m$ is the mixing parameter ranging from 0 (no mixing) to 1 (maximum mixing), and $z_{LCL}$ is the LCL calculated using only the $z_1$ station. Then, the LCL is iteratively calculated using $\theta_p$ and $q_p$, resulting from the combination of stations $z_1$ and $z_2$ in each iteration. To represent the topographic uplifting, we iterate the LCL calculation using Equation 3, with the number of iterations representing the degree to which land conditions ($z_2$) dominate marine conditions ($z_1$) and vice versa when Sc is advected inland.

The physical interpretation of this topographic uplifting is depicted in Figure 2b, where the initial iteration represents an equal influence of marine ($z_1$) and inland ($z_2$) conditions. Subsequent iterations represent the dominance of inland conditions over marine conditions. Our estimations show that the appropriate number of iterations is related to the distance in km between $z_1$ and $z_2$. Dominant marine conditions exhibit a higher latent heat flux (blue arrow in Fig. 2b) compared to sensible heat flux (orange arrow in Fig. 2b). Conversely, inland-dominant conditions showcase a prevalence of sensible heat fluxes over latent
heat fluxes (Fig. 2b). The shift in surface energy partitioning towards dominant sensible heat flux (inland conditions) leads to the LCL being reached at a higher altitude, resulting in the uplifting of the Sc cloud (Fig. 2c). This phenomenon is due to the warmer and drier conditions prevalent over land. It is important to note that during the advection of the Sc cloud, the MBL remains well-mixed, thereby minimizing differences between marine ($z_1$) and inland conditions ($z_2$).

To assess the accuracy of our CB estimations, Figure 4 presents a multi-temporal comparison between CB estimations
derived from the AMARU model and observations conducted in the Atacama Desert in 2017 as a part of the Ground Optical Fog Observations (GOFOS) experiment (del Río et al., 2021a). The GOFOS experiment entailed year-long monitoring of cloud base and top dynamics during an ENSO-neutral year (2017), employing optical cameras placed across the terrain to record the vertical movement of Sc cloud and fog. The left panel of Figure 4a illustrates that CB estimates generated by the model using Equation 3 (CB$_{mod(3)}$) closely align with those observed in 2017. The mean values of the estimated CB stand
at 879 m compared to the observed average of 870 m, with similar standard deviations of 88 m and 93 m, respectively. This satisfactory performance of the model in estimating CB is also observed at a monthly scale in Figure 4b, where the estimated CB generally differs by ∼50 m from the observed values on a monthly basis. To assess the model's capacity to replicate the





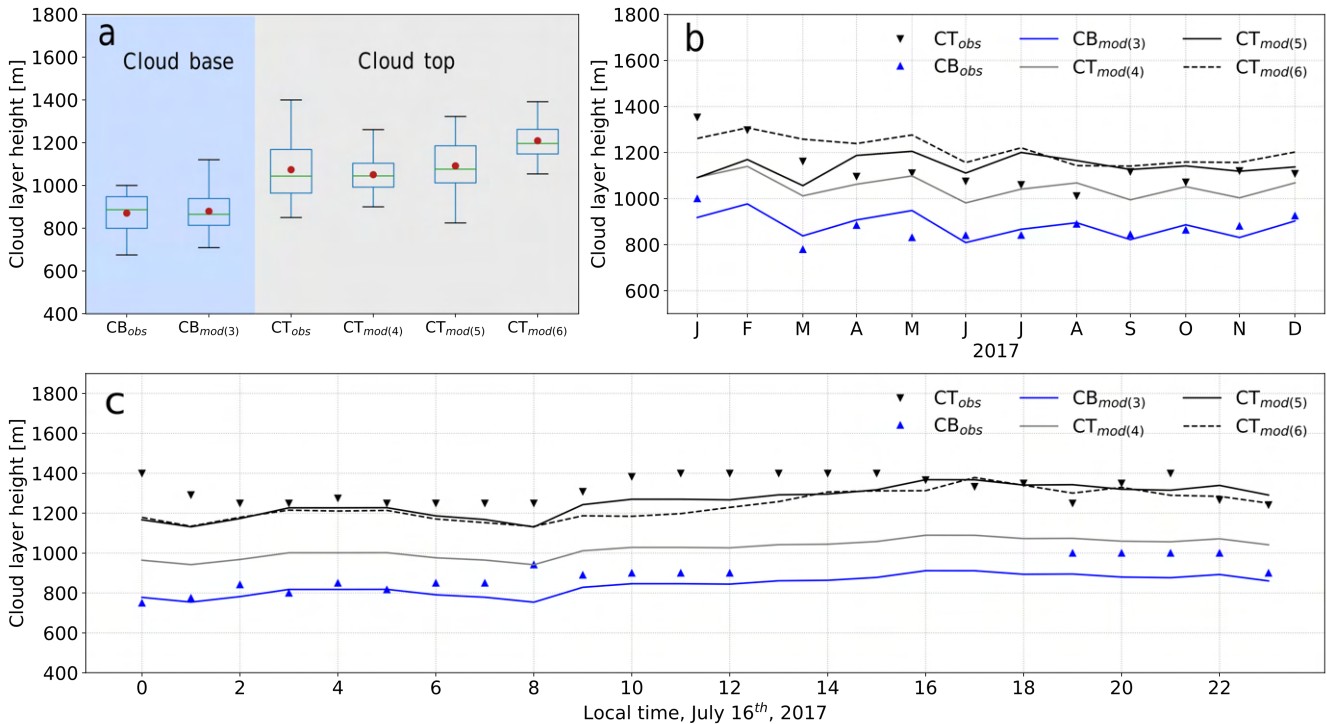

**Figure 4.** (a) Annual, (b) monthly-averaged, and (c) typical diurnal cycle of CB and CT comparisons between observations (obs) and modeling (mod). The number sub-fix refers to the equation number in sections 2.1.2 and 2.1.3.

diurnal cycle of CB, Figure 4c shows a representative foggy day in the Atacama region. It is evident from the figure that the estimated CB closely tracks its diurnal cycle, with errors of ∼100 m observed during the afternoon.

### 2.1.3 Cloud Top, CT

The parcel method, upon which our CB calculations are founded, determines $q_l$ from the LCL level upward, according to atmospheric pressure decreases. However, atmospheric pressure also decreases beyond the MBL, where the Sc stands. Consequently, it becomes necessary to estimate the Cloud Top (CT [m]) in order to calculate the $q_l$ within the cloud layer. Given the challenges associated with estimating CT using basic meteorological data and taking advantage of the homogeneity of Sc as a cloud layer, we propose estimating CT as a function of modeled CB using three simple linear regression models. These models are phenomenological expressions based on CT measurements obtained during the GOFOS experiment in 2017 (del Río et al., 2021a). The proposed linear regression models are as follows:

$$CT_{mod(4)} = 236.47 + 0.9355(CB) \qquad (4)$$





$$CT_{mod(5)} = CB + CB\sqrt{\frac{FCF}{2}} \qquad (5)$$


$$CT_{mod(6)} = 236.47 + 0.9355(CB)(1 - \frac{\partial \theta}{\partial z})100 \qquad (6)$$

Equation 4 shows a linear regression model in which CT [m] is solely dependent on CB [m]. Equations 5 and 6 correspond to linear regression models where CT is determined by CB and fog occurrence. The FCF (fog cover fraction [%], section 2.1.1.) in Equation 5 and the vertical thermal gradient ($\partial \theta / \partial z$ [K m$^{-1}$], Fig. 3a) in Equation 6 are based on observations conducted during the GOFOS experiment, where CT demonstrates a negative correlation with fog frequency (del Río et al., 2021a). A

comparable linear regression model, combining CB and fog occurrence to estimate CT, has been tested in various locations within the coastal Atacama region by Lobos-Roco et al. (2024).

Figure 4 shows the effectiveness of linear regression models in predicting CT compared to observations obtained from the GOFOS experiment. The right panel of Figure 4a shows the performance of the three linear regression models against observations for the year 2017. The annual means of the three models are similar to the observed value of (1073 m), with respective

values of 1050 m, 1091 m, and 1209 m. However, the CT derived from Equation 5 is the one that performs better, exhibiting a standard deviation of 142 m compared to the observed value of 124 m. At the monthly scale (Fig. 4b), the CT estimated by Equation 6 overestimates observations by 150 m. However, the CT from derived from equations 4 and 5 remains within a 50 m range of the observed values. In Figure 4c, showing a representative diurnal cycle during the foggy season, both observed and modeled CTs are presented. Here, it is evident that the CT estimated by equations 5 and 6 demonstrate a better performance,

closely aligning with observations (black triangles). However, the CT estimated from Equation 4 underestimates observations by over 200 m. These three linear regression models offer a statistical framework for estimating CT, with performance varying based on temporal scale. Henceforth, in this manuscript, we adopt the CT derived from Equation 5.

### 2.1.4 Liquid Water Content

Once we have estimated fog occurrence (FCF), the fog Cloud Base (CB), and Cloud Top (CT) using simple meteorological data

from a topographic transect, we proceed to determine the adiabatic liquid water content $q_l$ within the cloud layer (z: CT-CB). To achieve this, we utilize the following equation:

$$q_{l(z} = q_{(z)} - q_{s(z)}; q_l \geq 0 \qquad (7)$$



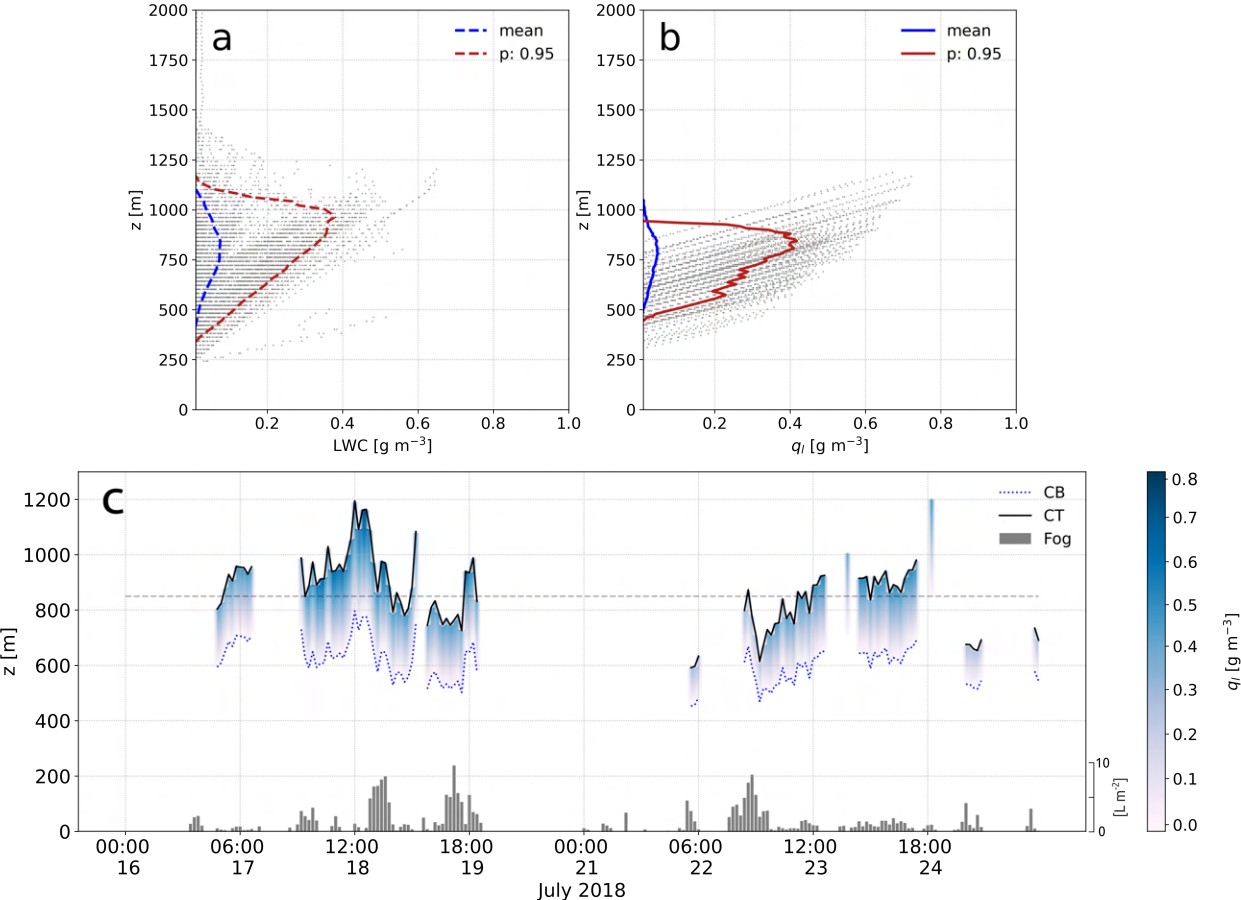

**Figure 5.** Vertical profiles $q_l$ (in g m$^{-3}$) (a) derived from a microwave radiometer and Doppler lidar observations (Schween et al., 2022; CLU, 2024) and (b) estimated by the AMARU model over Diego Aracena Airport. Grey dots represent hourly-averaged profiles filtered by percentile 0.99; red lines represent percentile 0.95, and the blue line represents the mean. (c) The evolution of the vertical profile simulated by the AMARU model for the Alto Patache Site. Grey bars at the bottom represent fog collection measurements at 850 m (dashed line).

where $q$ is the total specific humidity, $q_s$ denotes the saturated specific humidity, and represents the vertical level between CB and CT (Fig. 2a). It is important to note that, using a combination of $z_1$ and $z_2$ stations as in Equation 3, the term $q$ is replaced in each iteration by $q_{1,2}$. Regarding $q_s$, since this term is dependent on absolute temperature, it is influenced by $\theta(1,2)$.

Figures 5a and 5b show the validation of the model-estimated liquid water content ($q_l$) against observations (CLU, 2024) derived from combined measurements of a microwave radiometer with a Doppler lidar (Schween et al., 2022), conducted at the Diego Aracena Airport in the coastal Atacama Desert during July 2018. Our estimations of liquid water content obtained from Equation (7) systematically double the observed values. Consequently, we applied a correction factor of 0.5 to our estimations, as illustrated in Figure 5b.





In general terms, Figures 5a and 5b show a satisfactory comparison between our modeled estimations of liquid water content and the measured ones. The mean observed values peak at 0.1 g m$^{-3}$ at 800 m altitude, while our mean estimations peak at 0.08 g m$^{-3}$ at the same altitude ($\sim$800 m), consistent with typical values found in marine Sc clouds. Likewise, the 0.95 percentile curve (red line in Fig. 5) closely tracks the vertical distribution of observations, exhibiting peaks of 0.4 g m$^{-3}$

between 800 m (modeled) and 1000 m (observed). Upon comparing the maximum hourly-averaged values (grey dots) between the model and observations, we observe that the model overestimates observations by $\sim$0.1 g m$^{-3}$. Finally, upon integrating the vertical column of LWC, we observe significant similarities in the mean liquid water path (LWP) with values of 1.8 kg m$^{-2}$ and 1.6 kg m$^{-2}$ for modeled and observed data, respectively. To validate the results obtained from the thermodynamic module of the AMARU model, Figure 5c presents the temporal evolution of a simulated fog cloud during a fog event occurring

between July $16^{th}$ and $25^{th}$, 2018, at the Alto Patache site. The figure illustrates the model's capability to accurately represent fog-cloud frequency, its vertical structure, and water density ($q_l$) over time. In terms of fog frequency, our model shows fog-cloud formation from the $17^{th}$ to the $19^{th}$ and from the $22^{nd}$ to the $24^{th}$, aligning with the periods of highest fog collection rates (grey bars). From the $19^{th}$ to the $22^{nd}$, our model does not depict cloud formations, consistent with near-null fog water collection during this period. Likewise, we observe that changes in the vertical structure of the cloud (base and top) correspond

to variations in liquid water content and fog collection.

In summary, our straightforward methodology, employing a transect of meteorological stations, effectively estimates the $q_l$ within the MBL vertical column. This estimation is achieved by combining thermodynamic principles and statistical regressions, supported by climatological observations. Notably, our approach not only provides estimates of $q_l$ but also of fog frequency and the vertical structure of the fog cloud, thus enhancing our understanding of the fog phenomenon in coastal arid

regions.

### 2.2 Water potential module: Maximum Water Potential ($MWP$) and collector efficiency coefficient ($\eta$)

The second critical parameter in our proposed model is the collector efficiency coefficient ($\eta$). This variable is intricately linked with complex processes and factors such as wind flow, liquid water content, droplet size, collector positioning, material properties, mesh curvature and porosity (Carvajal et al., 2020). To ensure our assumptions align with climatological observations, we

determine the collector efficiency using an empirical coefficient. This coefficient is defined as the ratio between the Maximum Water Potential (MWP) and observed fog collection. As a ratio, $\eta$ represents the percentage of the maximum water that a fog collector can potentially capture under given atmospheric conditions. $\eta$ is calculated as follows:

$$\eta = \frac{f_{obs}}{MWP}, \tag{8}$$

where:





| Coordinate | Altitude | Time period | mean | 25% | 50% | 75% |
|---|---|---|---|---|---|---|
| 19.17° S- 70.17° W | 850 m | 2022 | 16% | 5% | 13% | 26% |
| 20.48° S- 70.05° W | 1200 m | 2019 | 26% | 6% | 16% | 30% |
| 20.82° S- 70.14° W | 850 m | 2018 | 36% | 18% | 34% | 51% |
| 30.65° S- 71.68° W | 630 m | 2022 | 27% | 6% | 20% | 45% |
| 32.16° S- 71.49° W | 650 m | 2022 | 15% | 4% | 5% | 21% |

**Table 1.** Descriptive statistics of the empirical efficiency coefficient $\eta$ estimated in five fog collection stations along a 2000 km coastal strip in Chile.

$$MWP = uq_lA \qquad (9)$$

and $f_{obs}$ represents the fog collection observations. Note that both $MWP$ and $f_{obs}$ are averaged per hour, therefore both terms have the unit of L m$^{-2}$ h$^{-1}$. Since $\eta$ is calculated based on fog observations, its value depends on the type of collector used, providing flexibility to the model to adapt to different collector types if observations are available.

Table 1 shows the empirical collector efficiency coefficient ($\eta$) calculated for five fog collection stations located between 600 and 1200 m along the coastal strip of Chile. Overall, $\eta$ varies from 15% to 36%, with a variability ranging from 4% to 50%. Three factors contribute to this variability in the efficiency coefficient. Firstly, the model's ability to accurately determine fog frequency (RMSE of 6% in Fig. 3a) can lead to discrepancies, potentially resulting in very high ($\eta \sim 100\%$) or null ($\eta \sim 0\%$) efficiencies when fog collection is observed, thus altering the averages. Secondly, wind speed may also play a significant role, as it is responsible for transporting $q_l$ through the collector. Lastly, both the material of the mesh and its curvature during fog collection could impact mesh efficiency (Carvajal et al., 2020). Despite the variability in $\eta$ across all sites, we find an average efficiency coefficient of 22% $\pm$ 10%, consistent with results in the literature. For instance, Montecinos et al. (2018) reported efficiencies ranging from 0% to 36% in large fog collectors. Similarly, using numerical simulations, Carvajal et al. (2020) reported a mean efficiency of 28% with a theoretical maximum of 36%. Finally, de Dios Rivera (2011) reported maximum fog collection efficiencies between 20% and 24% using a simple numerical model approach for different mesh types.

Once $\eta$ is estimated, we can readily solve Equation (2) to obtain an estimation of fog water harvesting ($q_h$). Given that $q_l$ has a vertical dimension, assuming a constant wind speed ($u$) along the MBL, we can derive the vertical distribution of fog harvesting.

## 2.3 Spatial module: fog harvesting maps

In addition to the thermodynamic module, we propose a spatial module for interpolating the vertical variability of $q_h$ into a horizontal spatial domain. To do it, we integrate the thermodynamic and water potential modules results with two satellite products: a digital elevation model (DEM) and GOES satellite images. We outline four steps to achieve this spatial variability.





The first step involves reclassifying the DEM pixels based on the cloud layer height, removing all pixels below the CB and above the CT. This reclassification ensures that only the altitude range where the Sc cloud could potentially impact the topography is considered. In the second step, we create an aspect image (slope orientation) with the DEM and reclassify the pixels based on the range angle of the main wind direction when fog is collected (obtained from observations at the $z_2$ station). The third step involves calculating the fog and low cloud (FLC) frequency using data from the GOES satellite (del Río et al., 2021b; Lobos-Roco et al., 2024). This algorithm continuously calculates the presence and absence of FCL in every pixel. The third step serves as a spatial framework, delineating the area where fog-cloud interaction with topography occurs. The spatial intersection of the three steps generates optimal areas for fog collection, physically representing the locations where the Sc cloud intersects with the surface. It is important to note that the values of pixels in these optimal areas for fog collection represent altitudes (m ASL). The final step involves replacing the altitude pixel values of the optimal fog collection areas with the vertical distribution of potential fog harvesting ($q_h$). As $q_h$ values are associated with a vertical domain ($z$), each $q_h$ value can be mapped onto the resulting grid of optimal fog collection areas. The result of this last step yields a spatial distribution of potential fog harvesting.

## 3   Model applications to (semi-)arid study case sites

The AMARU enables us to evaluate the spatiotemporal variability of fog harvesting using routine meteorological data and satellite products. In this section, we evaluate the application of the model ($q_h$) at three sites along the coastal strip of Chile corresponding to hyper-arid, arid, and semi-arid ecosystems between 2018 and 2023.

Figure 6 shows the geographical setting of the study sites, which correspond to hyperarid (Site a), arid (Site b), and semi-arid (Site c) fog ecosystems situated between 600 and 1200 m ASL along the coastal mountains of Chile. Generally, these sites represent xeric ecosystems (Muñoz-Schick et al., 2001) sustained year-round by fog, with a frequency exceeding 40% (Fig. 6). Each of these three sites is equipped with meteorological and fog collection observations, managed by the Centro UC Desierto de Atacama of Pontificia Universidad Católica de Chile. The characteristics of these stations and their data and parameters used in the model are summarized in Table 2. In addition, to meet the model's requirements, observations from these three sites ($z_2$) are complemented with data from near-sea level observations ($z_1$), sourced from public datasets (www.agromet.cl), which are also detailed in Table 2.

### 3.1   Seasonal cycle of modeled and observed fog harvesting

AMARU satisfactorily reproduces the observations of fog harvesting in both magnitude and variability over time. Figure 7 shows a comparison of monthly-average daily rates of fog harvesting at the three analyzed sites. Overall, the model results (blue) follow the seasonal cycle of observed fog collection (grey) across latitudes, albeit underestimating observations by 2% to 20%. In the hyperarid environment of Site (a) (Fig. 7a), the model estimates an annual daily rate of 4.1 L m$^{-2}$ d$^{-1}$, which satisfactorily compares to the rate of 5.5 L m$^{-2}$ d$^{-1}$ obtained through observations. Likewise, the model can closely track the seasonal cycle of fog harvesting, exhibiting low rates in summer (JFM) and autumn (AMJ) and higher rates in winter



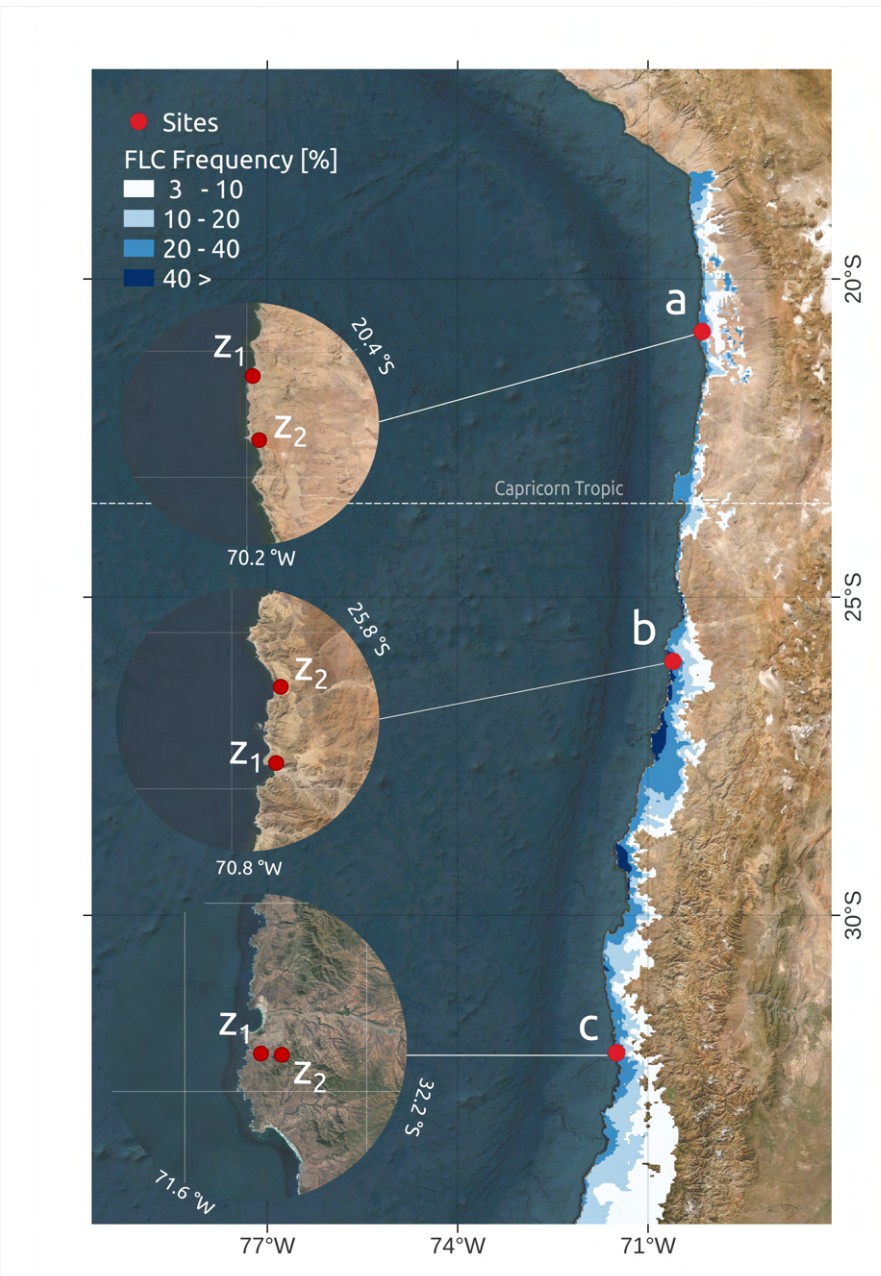

**Figure 6.** Location of the study sites and their meteorological stations. The blue-coloured areas represent the fog and low cloud (FLC) frequency obtained by the GOES satellite (del Río et al., 2021b) between 2018 and 2023. $z_1$ and $z_2$ represent the meteorological stations forming the transect used for running the model according to our methodology.





| Site | Coordinates | Height | Distance coast | $\eta$ | Available data | Time period |
|------|-------------|--------|----------------|--------|----------------|-------------|
| $a_{z_2}$ | 20.82° S-70.14° W | 850 m | 5 km | 36% | T, RH, P, U, WD, fog collection | 01-01-2018 |
| $a_{z_1}$ | 20.52° S-70.15° W | 48 m | 1 km | | T, RH, P, U, WD | 31-12-2018 |
| $b_{z_2}$ | 26.00° S-70.60° W | 820 m | 2 km | 22% | T, RH, P, U, WD, fog collection | 01-05-2023 |
| $b_{z_1}$ | 26.29° S-70.62° W | 120 m | 2 km | | T, RH, P, U, WD | 31-12-2023 |
| $c_{z_2}$ | 32.16° S-71.49° W | 650 m | 3 km | 22% | T, RH, P, U, WD, fog collection | 01-01-2022 |
| $c_{z_1}$ | 32.16° S-71.51° W | 60 m | 1 km | | T, RH, P, U, WD | 31-12-2022 |

**Table 2.** Geographic characteristics and available data of observational sites ($z_2$) and their corresponding stations at the coast ($z_1$). $T$ represents air temperature at 2 m, $RH$ relative humidity, $P$ air pressure, $U$ wind speed, and $WD$ wind direction.

(JAS) and spring (OND). Moreover, the model correctly estimates the frequency of fog events. For instance, during summer, the model estimates a very low (JM) or null (F) fog collection, with errors around 0.39 L m$^{-2}$ d$^{-1}$ (20%) compared to observations. Similarly, during the optimal fog harvesting season between winter and spring, the model correctly estimates the monthly magnitude of observed fog collection with errors of 2 L m$^{-2}$ d$^{-1}$ (18%). Finally, the model successfully replicates the variability in the monthly daily rates of fog collection as indicated by the error bars. In spring, for example, the observed rates range from 4 to 9 L m$^{-2}$ d$^{-1}$, while the model estimates rates range from 6 to 10 L m$^{-2}$ d$^{-1}$.

For Site (b), situated in an arid environment (Fig. 6), the amount of fog collection is notably lower compared to Site (a) (hyperarid). However, the model accurately reproduces the annual-averaged daily rate of fog harvesting with a slight underestimation of 2.6%. Despite this overall good performance, the model still underestimates observations by approximately 50% during winter-spring in terms of magnitude and variability (as indicated by the error bars). Unfortunately, the annual cycle for Site (b) remains incomplete as observations began only in May 2023. For the semi-arid environment of Site (c) the model shows annual daily rates of fog harvesting similar to those of Site (b), albeit with an overestimation of 5.8% compared to observations. During months with the lowest fog collection rates (March to July), the overestimation amounts to 1.8 L m$^{-2}$ d$^{-1}$. Conversely, during the months with the highest fog collection rates (August to December), the model underestimates observations by ~2 L m$^{-2}$ d$^{-1}$. It is worth mentioning that these discrepancies in estimation are not systematic, and despite them, the model captures the same seasonal cycle obtained through observations at all three sites.

## 3.2 Vertical variability of fog harvesting potential ($q_h$)

Since the model estimates the liquid water content ($q_l$) in the vertical column of the Sc cloud when it interacts with topography, and assuming constant wind at $z_2$ throughout the vertical, we can model the fog harvesting potential at every height within the Sc cloud layer.

Figure 8 shows the vertical variability of $q_h$ potential for the three analyzed sites. In the figure, dots represent the total $q_h$ per hour at each height within the fog-cloud layer over the course of one year. The red line depicts the annual average daily rate of $q_h$ as a function of height, while the black dot shows the observed annual average daily rate. In addition, the dots are colour-coded based on the corresponding $q_l$ values. From Figure 8, it is evident that fog harvesting potential decreases from

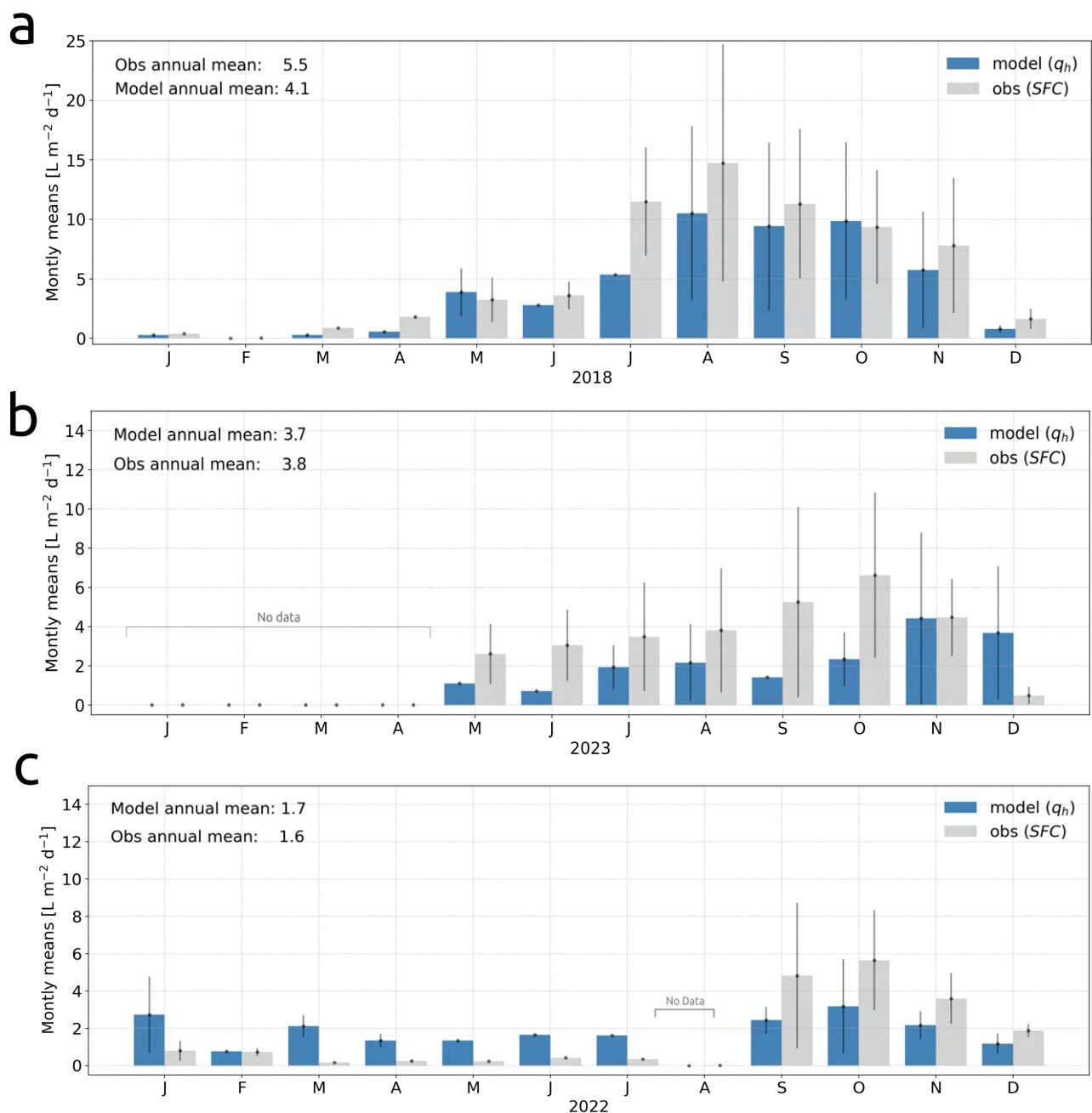

**Figure 7.** Comparison of monthly-averaged daily fog collection rates between model (blue) and observations (grey) in three fog ecosystems situated in the (a) hyperarid, (b) arid, and (c) semi-arid Chilean coast. The error bars show data variability between the $25^{th}$ and $75^{th}$ percentiles.





the hyperarid (north) to the semi-arid (south) regions for both $q_h$ and $q_l$. Specifically, in the hyperarid site, a $q_h$ of 10 L m$^{-2}$ d$^{-1}$ can be easily reached, whereas in the arid and semi-arid sites, maximum $q_h$ of 5 and 3 L m$^{-2}$ d$^{-1}$, respectively, are

observed. The same behaviour is observed for $q_l$, which exhibits higher values (mean percentile 0.95 up to 0.44 g kg$^{-1}$) in the hyperarid site compared to the arid and semi-arid sites, where percentile 0.95 percentile reaches up to 0.40 g kg$^{-1}$ and 0.33 g kg$^{-1}$, respectively. The vertical variability of $q_h$ also allows us to study the vertical liquid water capacity of the fog cloud. For instance, in the hyperarid site, the model estimates fog harvesting potential from 600 m up to 1350, while in the arid and semi-arid sites, fog can be harvested from 500 to 1250 and from 370 to 1050, respectively. These variations in $q_l$ and the fog

cloud layer height are explained in Equation (3) and Figures 2b and 2c. In Equation (3), we show that the calculation of cloud base, and consequently $q_l$, is influenced by the combined conditions of stations $z_1$ and $z_2$. For example, in the hyperarid site situated within the tropics (Fig. 6), air temperature is higher at both $z_1$ and $z_2$ compared to the semi-arid site. This implies that the condensation of the air parcel in Site (a) will occur at a higher altitude than in Site (b). Likewise, higher temperatures increase the air's capacity to hold humidity, resulting in a higher $q_l$ observed in the hyperarid site compared to the semi-arid

one. Another significant factor contributing to the difference in $q_l$ and cloud layer height is the distance from the coast at which station $z_2$ is located. For instance, the hyperarid site is 5 km inland compared to the arid and semi-arid sites located 2 km and 3 km away from the coast, respectively (Table 2). Consequently, inland conditions in the hyperarid site are hotter than in the other two sites, contributing to the formation of the cloud layer at higher altitudes.

Figure 8 also shows the annual average daily rates (red line) estimated by the model and observed by a standard fog collector

(black dot). This red line indicates the vertical placement of the maximum annual $q_h$. For example, in the hyperarid site, the maximum $q_h$ is located at 900 m in height, while observations are situated at 850 m ASL, explaining the highest annual daily fog collection rates. In contrast, in the arid and semi-arid sites, the maximum $q_h$ is not aligned with the height of the observations. For Site (b) case, the maximum $q_h$ is reached at ∼680 m, whereas observations are located at 820 m ASL. Similarly, for Site (c), the maximum $q_h$ is situated at 500 m, while observations are at 650 m ALS. The validation of annual

average daily rates in Figure 8 is determined by the proximity of the black dot to the red line at the observed height. For example, in Figure 8a, we observe an underestimation by the model, which is also evident in Fig 7a, but not in the vertical dimension, as the observations differ by ∼1.5 L m$^{-2}$ d$^{-1}$ from the modeling results. For sites (b) and (c) (Figs. 8b and 8c), the model accurately reproduces the annual daily rates, consistent with the observation, as also observed in Figures 7b and 7c.

### 3.3  Spatial variability of $q_h$: fog harvesting potential mapping

The combination of AMARU's results with satellite products enables us to interpolate the influence of the Sc cloud over land and its potential harvesting in space. This subsection introduces two examples of AMARU's results in spatial variability that can be utilized for fog ecosystem delimitation and water planning.

Figure 9a shows the optimal fog harvesting areas highlighted in red, corresponding to the region where the Sc cloud interacts with the Earth's surface. For Site (c), these areas are displayed near the summit of the coastal mountains, specifically ranging

from 370 m to 1050 m ASL (Fig. 8c). In addition, based on data from the meteorological Station ($z_2$) at this site, the fog cloud flux originates from the South and Southeast (110° to 300°), which is reflected in the model's depiction of the mountain slopes

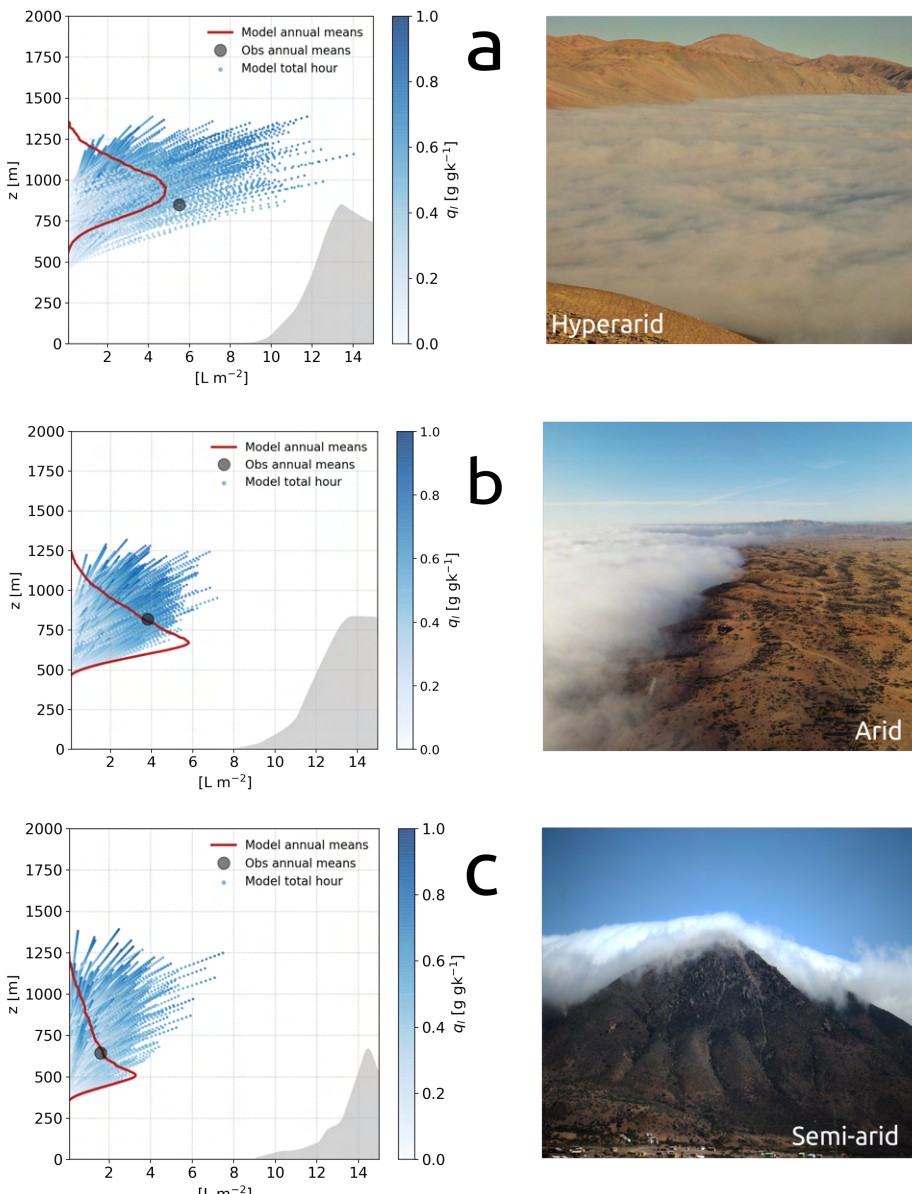

**Figure 8.** Vertical variability of modeled fog harvesting ($q_h$) in the (a) hyperarid, (b) arid, and (c) semi-arid sites. Dispersed dots represent the total fog harvesting at every hour; the red line is the annual average daily rate of $q_h$, and the grey dot is the observed annual averaged daily rate of $q_h$. The dispersed dots are color-coded in a blue scale representing the liquid water content ($q_l$). The grey shading represents the topographic profile of each site. The right panels show a photograph of each site during a fog event.





facing South and Southeast. In the zoom-out view of Figure 9a, we observe that the extent of these optimal fog harvesting areas spans within the first ∼20 km from the coast, as determined by the frequency of fog and low clouds derived from the GOES satellite (section 2.3).

To independently validate the spatial interpolation of AMARU's results, we compare the optimal fog harvesting areas with fog-dependant vegetation. In Figure 9a, areas highlighted in green represent the normalized difference vegetation index (NDVI) estimated through Sentinel satellite imagery. Overall, the optimal fog harvesting areas align with areas exhibiting the highest NDVI values. For example, a concentration of NDVI is observed at the summit of the mountains and the southeast slopes, indicative of a forest ecosystem sustained by fog (Garreaud et al., 2008). Furthermore, NDVI also concentrates at the bottom

of small valleys downstream of the summits, suggesting that fog water accumulated on the summits may potentially flow down, supplementing the precipitation input to the streams.

    Figure 9b shows the spatial variability modeled resulting from the intersection between the vertical profile annual average daily rate (red line, Fig. 8a) and the optimal fog harvesting areas. In the figure, we observe the spatial distribution of the fog water potential along the mountain, with maximum values observed around 900 m. The topography of the mountain favours

altitudes around 900 m ASL with Southwest slope orientations, leading the model to project large areas with fog harvesting potential ranging from 4 to 5 $\mathrm{L\,m^{-2}\,d^{-1}}$. In the eastern areas of the meteorological station (red dot, $z_2$), fog harvesting potential decreases to lower altitudes, consistent with results presented in Figure 8a. Likewise, decreases towards the southwest of the station at higher altitudes until it disappears. The area surrounding the station corresponds to the well-known fog oases of Alto Patache (Muñoz-Schick et al., 2001), situated between 600 and 850 m, within the optimal areas of fog harvesting determined

by the model.

    This model application is further extrapolated to the entire region to determine optimal fog harvesting spots within the area of influence of fog and low clouds, as determined through the GOES satellite. An example of these larger areas is shown in the zoom-out view of Figure 9b, where optimal fog harvesting areas are situated within 10 km from the coast. Since the model runs with simple meteorological time series, fog harvesting potential maps can be generated for different time averages, enabling

us to study and assess the spatial changes in fog harvesting potential over hours (events), days, seasons, and years.

## 4   Model limitations and challenges

Despite the versatility of the AMARU model in representing the harvesting of the advective fog phenomenon in both time and space, it has several limitations that are worth describing.

    Firstly, the model's capability to represent fog harvesting in time is primarily limited by the empirical collector coefficient.

However, this coefficient remains constant in the model, resulting in both underestimations and overestimations compared to observations. To improve our estimations of fog harvesting over time, further exploration into the collector efficiency is necessary, incorporating factors such as wind speed, collector material properties, and cloud droplet size into more complex functions.

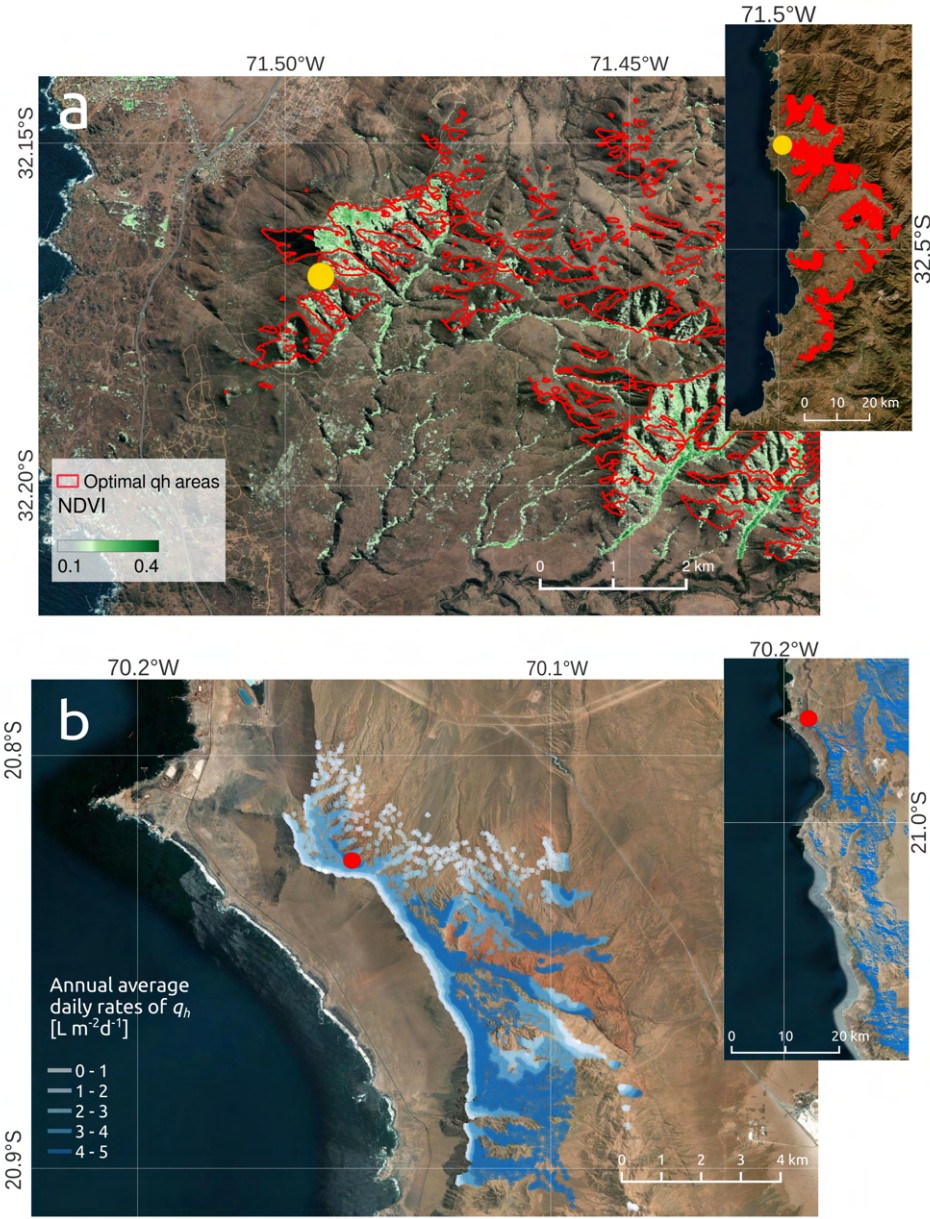

**Figure 9.** (a) optimal fog harvesting areas (red line) resulting from the model for Site (c) compared to the normalized difference vegetation index (NDVI, ranging from 0.1 to 0.4) estimated through Sentinel satellite images for 2022. The yellow dot indicates the meteorological station ($z_2$). (b) Spatial variability of annual averages daily rates of $q_h$ for the Site (a) estimated during 2018. The red dot indicates the meteorological station ($z_2$).





Secondly, the model's capability to assess fog harvesting potential in the vertical column of the MBL enables us to evaluate
the maximum fog harvesting potential beyond single-point observations. However, this vertical $q_h$ estimation is contingent
upon accurately determining $q_l$ assuming that wind speed remains consistent at every level of the MBL. Although $q_l$ estimations
align with observations from microwave radiometers, our results must be validated with in-situ observations of liquid water
content during fog collection. In addition, relevant physical processes influencing CT, such as dry air entrainment from the
free troposphere, among other are not included in its calculation. Instead, CT is statistically estimated, leading to uncertainties
in a variable whose precision is crucial for estimating the maximum $q_l$ and, consequently, $q_h$. Regarding wind speed, our
assumption of a constant horizontal wind along the MBL is based on the mixed layer theory, which posits that scalars such as
potential temperature, specific humidity, and wind speed remain constant if the MBL is well-mixed. However, this theory does
not consider topography, which may disturb this constant pattern when interacting with MBL winds. To improve the model's
estimation of $q_l$ and better $q_h$ potential, future research must incorporate accurate vertical profile observations of temperature,
humidity, and wind speed.

Finally, the spatial interpolation of $q_h$ represents a preliminary approach for fog harvesting potential mapping. This is because
its accuracy is limited by the availability of spatially distributed meteorological data. We spatially interpolate the conditions
determined by the model for the $z_2$ station to all surrounding areas that share the same geographic conditions. Nevertheless,
this approach may overestimate several inland locations that meet the geographical characteristics of $z_2$ but not the atmospheric
ones. Improving this spatial interpolation of $q_h$ can be addressed using two approaches. The first one involves utilizing gridded
meteorological data that allows us to solve Equation (2) at every grid point. Unfortunately, available gridded data is often too
coarse to accurately represent the sub-kilometer fog harvesting phenomenon. The second approach entails incorporating the
FLC frequency determined by the GOES satellite (Fig. 6) into the spatial interpolation of $q_h$. For example, we can modify $q_h$
spatially using a function based on the FLC frequency, where locations with similar geographical conditions to $z_2$ Station may
see their $q_h$ reduced (increased) if their FLC frequency is higher (lower) than that observed at $z_2$ Station.

## 5   Conclusions

We propose, formulate, and evaluate an observational-driven model, named AMARU, for estimating advective fog liquid water
capacity and potential harvesting in (semi-)arid regions. This model uses standard and routine meteorological observations,
enabling us to know where, when, and how much water can be potentially harvested from fog clouds.
The proposed model employs a thermodynamic approach to estimate the liquid water content of fog, incorporating key phys-
ical processes associated with the interaction between the stratocumulus cloud and topography. This approach yields vertical
profiles of liquid water content, from which fog frequency, cloud base, and top can be derived. In addition, by integrating the
estimations of liquid water content with climatological records of fog harvesting observations, we derive an empirical collector
efficiency coefficient to estimate vertical profiles of potential fog harvesting. Finally, combining vertical profiles of fog harvest-
ing potential with satellite products, we introduce a methodology for spatially interpolating these results, thereby generating
fog harvesting potential maps.





We apply the model to study the fog harvesting potential of three fog ecosystems situated along Chile's (semi-)arid coastal strip. Our results indicate that the AMARU model satisfactorily replicates the seasonal cycle of observations in terms of magnitude and variability, albeit with a slight underestimation (∼10%) of fog harvesting measurements. However, the model

accurately estimates the fog harvesting potential within the cloud vertical column across different arid ecosystems, revealing maximum potential harvesting rates exceeding $10 \, \text{L} \, \text{m}^{-2} \, \text{d}^{-1}$. Finally, we present two representative examples showcasing the application of the spatial interpolation derived from the model. The first delineates optimal fog harvesting areas, corresponding physically to the Earth's surface in contact with fog. The second determines the spatial distribution of modeled fog collection over complex topography, facilitating the mapping of fog harvesting water potential.

Given that the AMARU model relies on routine meteorological data, it can be used to study fog harvesting potential across different temporal dimensions: past, present, and future. For example, our model can complement numerical weather forecasting models to predict fog harvesting yields in the near future. Moreover, by using future climate data projections, it becomes feasible to assess how these water resources might respond to climate change.

Our model offers a versatile approach with multiple applications in massive fog harvesting planning, ecosystem delimitation

for conservation purposes, and the study of the climatological evolution of cloud water, among others. Given that fog is a global meteorological phenomenon, this model holds potential for worldwide applicability, thereby addressing data deficiencies in regions where fog harvesting represents a viable water source. We expect this research to yield significant social benefits by providing decision-makers with valuable insights into new water sources, thus aiding in the mitigation of climate change impacts.

*Data availability.* https://data.mendeley.com/datasets/c5s6zk2rmz/2

**Appendix A**

Starting from Equation (1), AMARU model defines fog harvesting ($q_h$) as the difference between total specific humidity input ($q_{in}$) and output ($q_{out}$):

$$\frac{dq_h}{dt} = q_{in} - q_{out}, \tag{A1}$$

Since the total specific humidity is transported into a collector by horizontal wind ($u_x$) (Fig. 1), we include it as follows:

$$\frac{dq_h}{dt} = u_x \frac{q_{in} - q_{out}}{\partial x}, \tag{A2}$$



Here, we set our first assumption, defining total specific humidity as $q = q_l + q_s$, where $q_l$ is the liquid water content and $q_s$ is the saturated specific humidity. Since $q_l$ two orders of magnitude lower than $q_s$, we assume that $q_{s(in)} \simeq q_{s(out)}$. This assumption allows us to cancel $q_s$ setting Equation (A2), in terms of $q_l$ as:

$$\frac{dq_h}{dt} = u_x \frac{q_{l(in)} - q_{l(out)}}{\partial x} \tag{A3}$$

Now, we introduce the collector area (Fig. 1) to Equation (A3) as:

$$\frac{dq_h}{dt} = u_x \frac{q_{l(in)} - q_{l(out)}}{\partial x} \partial z \partial y \partial x, \partial z \partial y = A \tag{A4}$$

Where $\partial x$ is canceled. Finally, we introduce our last assumption given by $q_{l(in)} > q_{l(out)} > q_h$, which allows us to state that $q_{l(out)}$ and $q_h$ are fractions of $q_{l(in)}$. This fraction $\eta$, represents the efficiency of the collector and can be estimated empirically through fog collection observations using Equation (8). By introducing $\eta$ in Equation (A4) the final model equation reads:

$$\frac{dq_h}{dt} = u q_l A \eta, \tag{A5}$$

*Author contributions.* **Felipe Lobos-Roco**: Conceptualization, Methodology, Software, Writing-Original draft preparation, Data curation, Visualization, Formal Analysis, Writing-Reviewing and Editing, Funding acquisition. **Jordi Vilà Gerau-de Arellano**: Conceptualization, Formal Analysis, Writing-Original draft preparation, Writing-Reviewing and Editing. **Camilo del Río**: Investigation, Conceptualization, and Resources.

*Competing interests.* The authors declare that they have no conflict of interest

*Acknowledgements.* This research was funded by CMPC contract n. 6496162. We acknowledge Centro UC Desierto de Atacama for providing data, discussions, and pictures. FL-R acknowledges FONDECYT project no. 1211846 for providing valuable data. Likewise, we acknowledge Cristobal Merino, Valentina Pacheco, Sebastian Vicuña, and Diego Ibarra for their help in standardizing databases. Moreover, we thank Nicolas Valdivia for photograph B in Figure 8 and the web site: https://www.davidnoticias.cl/cerro-santa-ines-los-vilos-se-declara-santuario-la-naturaleza/ for photograph C in Figure 8. Finally, we acknowledge E. Fiorin for her English language editing.



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
