# Peer review of "Observation-driven model for calculating water harvesting potential from advective fog in (semi-)arid coastal regions"

_Hydrology and Earth System Sciences, 2024_

## Referee Comment (RC1)

Review of HESS 2024-110

Observation-driven model for calculating water harvesting potential from advective fog in (semi-)arid coastal regions. By Felipe Lobos-Roco, Jordi Vilà-Guerau de Arellano, and Camilo del Río.

General comments.

Estimating water harvesting potential from advective fog in (semi-)arid coastal regions is an important subject and well worth the effort of developing and evaluating a model. I do however have serious reservations about the building blocks in the model, both Appendix A and Section 2.1.2. In the fog collector model, the variable definitions are not clear and, as I see it, mixing ratios would better that specific humidity in this context. For the cloud base and related parameters it seems extremely optimistic to determine cloud base from near surface (1.5 m?) measurements of T and q at two stations and used to infer the vertical gradients. In the end a factor 0.5 is applied to the results (line 219).

I would like to see the model details cleaned up and more clearly explained

Detailed comments

line 75: "total specific humidity (q)". Maybe make it clear that this is mass of water vapor + droplets + (if present) ice/Total mass of a parcel, and presumably in kg/kg. Many authors will use q as a symbol for mixing ratio which in the atmospheric context of mass of water/mass of dry air.

line 79: Equation (1) appears to be dimensionally incorrect unless $q_h$ is defined as a time integral of q with units like kgs/kg? Give a clear definition of "fog harvesting capacity, $q_h$". Same with $q_{in}$ and $q_{out}$ - are these amounts the inflow and outflow specific humidities. Same concerns in Appendix A

line 83: $q_{in}$, $q_{out}$ and $q_h$ appear to have same dimensions?

line 83: Assuming $q_s(in) \simeq q_s(out)$ implies no temperature change including any release of latent heat due to water vapor condensation on the grid. Will that be true? The rationale used, "given that $q_l$ is by two orders of magnitude lower than $q_s$", is invalid. What matters is the size of $q_l$ relative to any change in $q_s$!

line 89, 90 and Appendix A: Figure1 is useful, and makes it clear that $q_h$ is NOT a specific humidity. Better to use a different symbol, and specify the units (maybe in kg rather than L/m$^2$). See notes at end.

line 122: How close to the shore is Fiego Aracina airport?

Figure 3: Points 10,11,12 seem to be missing. Negative correlations?

lines 134-139: I found this confusing. What time of day are the fog situations we are looking at? Earlier in the paper, sea breezes are cited as the cause of the flow from the ocean, so daytime, while here I am not sure. Does "stratified" mean stably stratified? Over water in the marine boundary layer solar radiation has only a small impact on sea surface temperature.

line 143+: 2.1.2 Cloud Base, CB It seems extremely optimistic to determine cloud base from near surface (1.5 m?) measurements of T and q at two stations used to infer the vertical gradients. The explanation is rather vague (Lines 160-170). The interpretation of the results seems rather generous. Is there any chance of some more comparisons with sounding data of some sort?

line 153: Equation (3) is probably wrong. Equations 3 and 4 in Lobos-Roco et al (2018) make sense, this does not. Also it is not clear to me exactly what subscript 1,2 means, and the iteration process needs more explanation.

line 208+: There is a missing ) in Eq (7) but it seems reasonable, Thus I was surprised to read (line 219) that " Our estimations of liquid water content obtained from Equation (7) systematically double the observed values. Consequently, we applied a correction factor of 0.5 to our estimations, as illustrated in Figure 5b. Something is wrong!

line 241+: Collection efficiency. Could this be determined in a laboratory wind tunnel study? Here it just looks like model tuning.

Fig 7b: Annual means seem wrong, based on monthly values shown.

Alternative analysis.

1) Use q, $q_l$, $q_s$ as mixing ratios rather than specific humidity - since, with specific humidity, mass of total parcel will change from $q_{in}$ to $q_{out}$.

[Figure]

Is the assumption that the air is always saturated, so that water vapor mixing ratio $q_v = q_s(T)$, necessary?

Work with fluxes of **liquid water**, and let A = dydz

Flux in, $Fl_{in} = \rho_a u A q_{lin}$ ; Flux out, $Fl_{out} = \rho_a u A q_{lout}$, both in kg/s,

Then water collection rate, $Q_h = Fl_{in} - Fl_{out} = \eta$ Adx $Fl_{in}$ , also in kg/s with $\eta$ having dimensions m⁻³. With a different symbol, $Q_h$, since this is different from other q usage.

This has assumed no vapor-liquid transfers as the parcel goes through the mesh. But is that true? Is there "dew" produced as well as cloud drops captured?

---

## Referee Comment (RC2)

HESS submission by Roco et al

This work uses a simple 1D advective fog model based on various assumptions for a given budget equation. It is claim that using satellite observations, model results and observations from MWR and adiabatic method are comparable, and they are very close to each other for annual representation. The work claims results can also be used for climatic fog research.

Based on my review, there are several errors existing in their work.

- This starts with equations and follows up with results. For example, mass balance equation is wrong, and assumptions are not presented or mentioned properly. What are they?
- Introduction is given in a large parag that doesn't focus on fog physics/dynamics etc at all. There are several works on marine fog (Gultepe et al 2021 BLM; 2019 Marine fog review; Fernando et al 2021) that are not mentioned. Characteristics of LWC, Nd, and DSD are provided in these works.
- Fog device for LWC is being used since 2006 (Gultepe et al BAMS and others).
- Where is the importance of Nd in the model? Without Nd, how do you get accurate LWC? What is physically missing here? What is the role of Nd in LWC?
- See for budget equations given for cirrus clouds in PAAG 1990s Gultepe et al where steady state assumption is assumed to solve hor and ver advections. In addition, what role turbulence, radiation, and vertical advection play role? What happened to these terms?
- Fig. 7 suggests that there are huge diffs between observed and model simulations per month, how can be annual values get closer so much? Something is wrong.
- How do you use satellite obs is not clear, how do you get fog LWC/LWP or coverage, no method is given properly
- Apply these results for climate is very much simplification, this should be taken out.
- Conclusions; needs to be collected for a few bullets and explained based on the text, not clearly explained properly.
- Appendix also has severe issues without providing assumptions.

Overall, I cant accept this paper as scientifically meaningful and it needs lots of work to be published.

---

## Author Comment (AC1)

**Reply to reviewer**

**Review of HESS 2024-110**

Observation-driven model for calculating water harvesting potential from advective fog in (semi-)arid coastal regions. By Felipe Lobos-Roco, Jordi Vilà-Guerau de Arellano, and Camilo del Río.

We appreciate the Reviewer's comments, suggestions, and alternative analysis, which have enabled us to improve our model formulation and, consequently, the manuscript. We have accepted all the Reviewers' comments, corrections, and suggestions, resulting in significant changes to the manuscript. Below, in blue font, you will find our responses to each comment, correction, and suggestion. We have also included the line numbers where changes were made in the revised version of the manuscript (Revised_manuscript.pdf).

**General comments.**

Estimating water harvesting potential from advective fog in (semi-)arid coastal regions is an important subject and well worth the effort of developing and evaluating a model. I do however have serious reservations about the building blocks in the model, both Appendix A and Section 2.1.2. In the fog collector model, the variable definitions are not clear and, as I see it, mixing ratios would better that specific humidity in this context. For the cloud base and related parameters it seems extremely optimistic to determine cloud base from near surface (1.5 m?) measurements of T and q at two stations and used to infer the vertical gradients. In the end a factor 0.5 is applied to the results (line 219).

I would like to see the model details cleaned up and more clearly explained

We have accepted most of the Reviewer's comments and alternative analysis for improving the model formulation, including detailed explanations of each variable, unit, and dimension. These revisions have resulted in major changes to the manuscript, particularly in Section 2 (model formulation and evaluation), Figure 1, Section 2.1, Section 2.1.2, Figure 2, Section 2.1.4, Figure 5, Figure 7, and Figure 8.

The main modifications involved the model formulation and assumptions. The primary difference in the revised model formulation is the inflow and outflow fog water, now expressed in terms of the mixing ratio r. This is thoroughly explained in Section 2. Additionally, Appendix A has been removed, and a new Figure 1 with updated definitions has been included. The variable '$q_h$' has been replaced by "$W_h$", which denotes water harvesting. Finally, the variable liquid water content ($q_l$) has been replaced with the adiabatic liquid mixing ratio ($r_l$), clearly defined as g of liquid water per kg of dry air. We have also rechecked the calculations for the adiabatic liquid mixing ratio, which are now more physically consistent and overestimate by 28% (0.2 g m$^{-3}$) when compared to the cloud radar. Despite the our estimations of $r_l$ are slightly higher than cloud radar observations (Fig. 5a and b), now they are more consistent and does not include any correction factor.

The new model formulation, included between lines 72 and 115 of the revised manuscript, reads as follows:

"The AMARU aims to estimate the liquid water content of Sc clouds and the potential for fog harvesting. Our goal is to formulate a model that can use the available routine meteorological data in an area with significant ocean-land contrasts and very complex topography. Figure 1 illustrates the physical assumptions and processes along with the respective variables and units. The model relies on an adaptation of the mass conservation equation. The sequence of physical mechanisms involved is as follows: during a fog event, a certain amount of liquid water ($W_h$) is retained from the total fog

inflow when passing through a passive collector. We assume that the harvested fog water results from the difference between fog inflow ($F_{in}$) and outflow ($F_{out}$) in g kg$^{-1}$ m s$^{-1}$. This equation reads as follows:

$$W_h \cong F_{in} - F_{out} \qquad (1)$$

Here, fog inflow and outflow are described as:

$$F_{in} = r_l\, u_x \qquad (2)$$

$$F_{out} = F_{in}(1 - \eta) \qquad (3)$$

where $r_l$ is the liquid water mixing ratio, defined as the amount of liquid water ($m_l$ in Fig.1) per unit mass of dry air ($m_d$) that contains it, expressed in g of water per kg of dry air. To calculate the inflow we use $u_x$, which represents the perpendicular wind speed (m s$^{-1}$) relative to the collector. The term $\eta$ is a dimensionless ration representing the collector efficiency. This coefficient is described as:

$$\eta = \frac{W_h}{F_{in}} \qquad (4)$$

where $\eta$ corresponds to the percentage of water harvested over the total water that can potentially pass through the collector (details in Section 2.2). Reordering the terms, we express Equation (1) in net terms as:

$$W_h = r_l\, u_x\, \eta \qquad (5)$$

The $W_h$ units are then g kg$^{-1}$ m s$^{-1}$. However, for operationalization, the units of $W_h$ are converted in L m$^{-2}$s$^{-1}$ (equivalent to mm) once grams are transformed to liters and dry air density (kg m$^{-3}$) is included as:

$$W_h = r_l\, \rho_a\, u_x\, \eta \qquad (6)$$

Finally, $W_h$ is integrated over a period as:

$$\overline{W}^{\Delta t}{}_h = \int_{t0}^{t1} W_h\, \partial t \, , \qquad (7)$$

where $t_0$ and $t_1$ correspond to the initial and ending time intervals. The model has three main assumptions described as follows: (1) $F_{in} > F_{out}$; (2) since the model aims to reproduce advective fog collection, it is assumed that condensation only occurs in the atmosphere under the conditions $r_l = r_v - r_s$; (3) the mixing ratio ($r_v$) being two orders of magnitude higher than $r_l$, is nearly conserved.

In Equation (6), $r_l$ and $\eta$ depend on location and condensation processes. Regarding location, $r_l$ varies in height (the vertical dimension of the model) and depends on the conditions of the marine Sc cloud over the ocean and its interaction with the topography. The second term, $\eta$ is related to cloud microphysics and the design and material properties of the collector. To delve into the details of $r_l$ and $\eta$, we break down the analysis of Equation (6) into two parts: the thermodynamic and water

potential modules. In addition, we introduce a third module for representing the model's horizontal spatial variability of $W_h$ trough spatial interpolation creating a fog harvesting potential map."

Regarding the cloud base, we understand the Reviewer's concerns about its estimation. However, we have thoroughly evaluated this estimation. To achieve this, we compared independent cloud base measurements from ground-based experiments (Fig. 4) and the observations obtained by a cloud radar (Fig. 5a and b) against our cloud base estimations. The comparison shows a high degree of agreement (Figures 4a and 4b, line 206), with errors ranging between 10 and 50 m. To clarify this, we have revised the manuscript to include detailed explanations of our calculation methods and the variables involved. We have also made some corrections to Equation (8) and Figure 2. These modifications, included between lines 171 and 183, are as follows:

"The second approach considers two physical processes involved in the Sc-to-fog transition: environmental mixing and topographic uplifting. Firstly, to represent the mixing with the environment experienced by an air parcel during adiabatic ascent, and based on Lobos-Roco et al. (2018), we combined the meteorological conditions measured at both transect stations ($z_1$, $z_2$) using a mixing parameter m as follows:

$$\varphi^p_{(z)} = (1 - m\frac{z}{z_{LCL}})\varphi^s + (1 - m\frac{z}{z_{LCL}})\varphi^{ML} \tag{8}$$

Where $\varphi$ is a scalar for potential temperature (θ) or specific humidity (q), super script p represents the state of the air parcel, s indicates the conditions at the lowest station used ($z_1$), ML refers to mixed-layer, which is an average of conditions observed at the two stations, m is the mixing parameter ranging from 0 (no mixing) to 1 (maximum mixing), and $z_{LCL}$ is the height at which LCL is reached. Secondly, to account for the inland effect (observed at $z_2$ station), LCL is calculated iteratively using an averaged θ and q ($\varphi^{ML}$) from $z_1$ and $z_2$. This $\varphi^{ML}$ and LCL are used in Equation 8 to estimate the air parcel state $\varphi^p_{(z)}$, which is then used to calculate a new LCL. This calculation is repeated several times, with $\varphi^{ML}$ being re-averaged with the conditions at station $z_2$ in each iteration. This repetitive calculation ensures that the inland conditions ($z_2$) in the MBL's state are accurately represented. Our estimations show that the appropriate number of iterations is related to the distance in km between $z_1$ and $z_2$,. For example, if $z_1$ and $z_2$ are separated by 5 km, we iterate five times."

**Detailed comments**

line 75: "total specific humidity (q)". Maybe make it clear that this is mass of water vapor + droplets + (if present) ice/Total mass of a parcel, and presumably in kg/kg. Many authors will use q as a symbol for mixing ratio which in the atmospheric context of mass of water/mass of dry air.

We agree with the Reviewer's comment., Please find our answer in the general comment section.

line 79: Equation (1) appears to be dimensionally incorrect unless q h is defined as a time integral of q with units like kgs/kg? Give a clear definition of "fog harvesting capacity, qh". Same with q in and qout - are these amounts the inflow and outflow specific humidities. Same concerns in Appendix A

line 83: qin , qout and q h appear to have same dimensions?

We agree with the Reviewer's comment, Please find our answer at the general comment section.

line 83: Assuming qs(in) ≃ qs(out) implies no temperature change including any release of latent heat due to water vapor condensation on the grid. Will that be true? The rationale used, "given that ql is by two orders of magnitude lower than qs", is invalid. What matters is the size of ql relative to any change in qs!

line 89, 90 and Appendix A: Figure1 is useful, and makes it clear that qh is NOT a specific humidity. Better to use a different symbol, and specify the units (maybe in kg rather than L/m 2 ). See notes at end.

We agree with the Reviewer's comment. Please find our answer at the general comment section, where we clarified our assumptions (line 104) and units (line 81,90, 99).

line 122: How close to the shore is Diego Aracena airport?

Diego Aracena Airport is very close to the shore, situated 0.42 km away and at an elevation of 48 m ASL. We have included this information in line 143 of the revised manuscript.

Figure 3: Points 10,11,12 seem to be missing. Negative correlations?

Yes, they are slightly to the left and behind the Y-axis line of the plot , indicating a negative correlation. We have added a short explanation about this negative correlation in the caption of Figure 3 as follows:

"Note that numbers 11 and 12 have a slightly negative correlation, placing behind the line and to the left of the Y-axis."

lines 134-139: I found this confusing. What time of day are the fog situations we are looking at? Earlier in the paper, sea breezes are cited as the cause of the flow from the ocean, so daytime, while here I am not sure. Does "stratified" mean stably stratified? Over water in the marine boundary layer solar radiation has only a small impact on sea surface temperature.

Lines 134-13 explain the relationship between MBL regimes and fog formation. We have based and connected our discussion to the findings of Lobos-Roco et al. (2018). In short, when we discuss the MBL, we refer to the conditions observed at the coast. This is because our observations ($z_1$, $z_2$) are taken over the continent. These observations enable us to quantify the advection of moist air that originates mostly during the day due to the ocean breeze. During the night, fog mostly dissipates as the MBL becomes stably stratified. However, our analysis shows that thermal and moisture vertical gradients follow different patterns. For example, Fig. 3a shows that orange (thermal) and purple (moisture) criteria are not similarly correlated with observations; thermal criteria show better correlations than moisture ones. This is explained by the coast's aridity, which, on the one hand, contributes in changing MBL thermal conditions, for example, stratifying the MBL during the night resulting in low fog frequency. On the other hand, these arid slopes do not contribute to moistening the MBL, with or without fog presence, resulting in low or even negative correlations with the fog frequency.

Yes, it means stably stratified, as opposed to well-mixed. We have enhanced this explanation by adding some key words to improve the text readability in lines 150- 160.

line 143+: 2.1.2 Cloud Base, CB It seems extremely optimistic to determine cloud base from near surface (1.5 m?) measurements of T and q at two stations used to infer the vertical gradients. The explanation is rather vague (Lines 160-170). The interpretation of the results seems rather generous. Is there any chance of some more comparisons with sounding data of some sort?

Unfortunately, no sounding data are available in the nearby area. However, we have contrasted our data with the ground-based observation detailed in Figure 4. For further clarification on our cloud base estimation, please refer to our answer to the general comment above.

line 153: Equation (3) is probably wrong. Equations 3 and 4 in Lobos-Roco et al (2018) make sense, this does not. Also it is not clear to me exactly what subscript 1,2 means, and the iteration process needs more explanation.

We agree with the Reviewer's comment, We have made significant changes in Section 2.1.2. Please refer to our answer to the general comment above.

line 208+: There is a missing ) in Eq (7) but it seems reasonable, Thus I was surprised to read (line 219)

that " Our estimations of liquid water content obtained from Equation (7) systematically double the observed values. Consequently, we applied a correction factor of 0.5 to our estimations, as illustrated in Figure 5b. Something is wrong!

We agree with the Reviewer's comment. The ")" was added. Moreover, we have introduced major changes in the model formulation, specifically in Section 2 (lines 72-115) and 2.1 (117-128; 171-190; 242-261). Please refer to our response to the general comment above for further details.

line 241+: Collection efficiency. Could this be determined in a laboratory wind tunnel study? Here it just looks like model tuning.

The collection efficiency factor is now better described (see our answer to the general comment above). This efficiency was not determined by a wind tunnel. Instead, we have estimated it using fog collector observations at several locations along the Chilean coast, though equation 13 (revised_manuscript). In our model runs, we use an efficiency ranging between 0.22 to 0.25, consistent with values reported in models and in-situ observations by Carvajal et al. (2020), Montecinos et al. (2018) and de Dios Rivera (2011) (lines: 295-300).

Fig 7b: Annual means seem wrong, based on monthly values shown.

We thank the Reviewer for her/his comment. The annual means refers to the averages of monthly daily rates (L m$^{-2}$ day$^{-1}$). For example, In the hyperarid site (Fig. 7a), monthly daily rates range from ~0 to ~10 L m$^{-2}$ d$^{-1}$, whit an annual average of 5. With the new model experiments, these values have been revised meticulously. In addition, we have revised the observations used for the model comparison, excluding those with low data quality during several months where the SFC measurements were affected by a broken mesh. These low-quality data were excluded from the new analysis. Consequently, "no data" labels appear in Figures 7b and 7c.

**Alternative analysis**.

1) Use q, q l , q s as mixing ratios rather than specific humidity - since, with specific humidity, mass of total parcel will change from q in to q out .

Is the assumption that the air is always saturated, so that water vapor mixing ratio $q_v = q_s(T)$, necessary?

Work with fluxes of liquid water, and let $A = dydz$

Flux in, $Fl_{in} = \rho_a uAq_{lin}$ ; Flux out, $Fl_{out} = \rho_a uAq_{lout}$ , both in kg/s,

Then water collection rate, $Q_h = Fl_{in} - Fl_{out} = \eta Adx Fl_{in}$ , also in kg/s with $\eta$ having dimensions m$^{-3}$. With a different symbol, $Q_h$ , since this is different from other q usage.

This has assumed no vapor-liquid transfers as the parcel goes through the mesh. But is that true? Is there "dew" produced as well as cloud drops captured?

We thank the Reviewer for the alternative analysis, which significantly contributed to improving our model formulation. For further details, please refer to our answer in the general comment section.

---

## Author Comment (AC2)

**Reply to reviewer#2**

HESS submission by Roco et al

This work uses a simple 1D advective fog model based on various assumptions for a given budget equation. It is claim that using satellite observations, model results and observations from MWR and adiabatic method are comparable, and they are very close to each other for annual representation. The work claims results can also be used for climatic fog research.

We appreciate the reviewer's comments, and based on them, we have introduced modifications to the model formulation and explanation of assumptions. Our model reproduces fog harvesting volumes using standard fog collectors. To this end, we have two goals: (1) to understand which processes govern the fog physics in the Atacama region; and (2) to optimize the use of routine meteorological data collected in regions with complex terrain to calculate water estimations. For goal 1, the main assumption is the simplification of microphysical processes. Here, we are aware that such processes are essential in the transition of low marine clouds and land fog formation. However, to justify these simplification, we have used two meteorological stations over a topographic transect facing the ocean to estimate the adiabatic liquid water mixing ratio (all or nothing) and its potential harvesting. In that respect, we would like to stress that the model has been systematically evaluated with the available observations, obtaining a reliable comparison. This comparison proves that, despite the model's simplicity, it reproduces the main characteristics of the fog over land physics in terms of fog frequency, its relation to the cloud base and top height, liquid water mixing ratio, and water harvesting.

Following the reviewer's comments, we have revised the entire model formulation, emphasizing on explaining the model assumptions. Regarding the model limitations, we have now discussed in more depth in section 4 and conclusions the purpose of the model. We place emphasis in describing the key role played by the available routine meteorological data to justify assumptions and to evaluate the mode. This last part is completed with a discussion on the model limitations and potential model improvements. In addition, we have added the suggestions and references provided by the reviewer in the introduction. Below, in blue font, you will find our responses to each comment. We have also included the line numbers where changes were made in the revised version of the manuscript (Revised_manuscript.pdf).

Based on my review, there are several errors existing in their work.

• This starts with equations and follows up with results. For example, mass balance equation is wrong, and assumptions are not presented or mentioned properly. What are they?

Based on comments provided by reviewers #1 and #2, we have introduced major changes in model formulation, including detailed explanations of each variable, unit, and dimension. In addition, we have elaborated more on the physical implications of the model assumptions. These revisions have resulted in major changes to the manuscript, particularly in Section 2 (model formulation and evaluation), Figure 1, Section 2.1, Section 2.1.2, Figure 2, Section 2.1.4, Figure 5, Figure 7, and Figure 8. The main changes in section 2 relating to formulation and assumptions are as follows:

The main modifications involved the model formulation and assumptions. The primary difference in the revised model formulation is the inflow and outflow fog water, now expressed in terms of the mixing ratio ($r$). This is thoroughly explained in Section 2. Additionally, Appendix A has been removed, and a new Figure 1 with updated definitions has been included. The variable '$q_h$' has been replaced by "$W_h$", which denotes water harvesting. Finally, the variable liquid water content ($q_l$) has been changed with the adiabatic liquid mixing ratio ($r_l$), clearly defined as grams of liquid water per kilograms of dry air. We have also double checked the calculations for the adiabatic liquid mixing ratio

to improve on the evaluation of the fog harvesting model. The new calculations are now more physically consistent and overestimate by 28% (0.2 g m$^{-3}$) when compared to the cloud radar. Despite our estimations of $r_l$ are higher than cloud radar observations (Fig. 5a and b), now they are more physical consistent since we can omit the correction factor included in the first version of the manuscript.

The new model formulation, included between lines 72 and 115 of the revised manuscript, reads as follows:

"The AMARU aims to estimate in a simple way the adiabatic liquid water content of Sc clouds and the potential for fog harvesting. Our goal is to design a model that use the available routine meteorological observations in an area with significant ocean-land contrasts and very complex topography. Figure 1 shows the physical assumptions and processes along with the respective variables and units. The model is derived from the mass conservation equation. The sequence of physical mechanisms are: (i) during a fog event, a certain amount of liquid water ($W_h$) is retained from the total fog inflow when passing through a passive collector. We assume that the harvested fog water results from the difference between fog inflow ($F_{in}$) and outflow ($F_{out}$) in g kg$^{-1}$ m s$^{-1}$. This equation reads as follows:

$$W_h \approx F_{in} - F_{out} \qquad (1)$$

(ii) Fog inflow and outflow are described as fluxes of the mixing ratio:

$$F_{in} = r_l \, u_x \qquad (2)$$

$$F_{out} = F_{in}(1 - \eta) \qquad (3)$$

where $r_l$ is the liquid water mixing ratio, defined as the amount of liquid water ($m_l$ in Fig.1) per unit mass of dry air ($m_d$) that contains it, expressed in grams of water per kilograms of dry air. To calculate the inflow we use $u_x$, which represents the perpendicular (mean ± std) wind speed (m s$^{-1}$) relative to the collector. (iii) The term $\eta$ is a dimensionless ration representing the collector efficiency. This coefficient is described as:

$$\eta = \frac{W_h}{F_{in}} \qquad (4)$$

where $\eta$ corresponds to the percentage of water harvested over the total water that can potentially pass through the collector (calculation in Section 2.2). Reordering the terms, we express Equation (1) in net terms as:

$$W_h = r_l \, u_x \, \eta \qquad (5)$$

The $W_h$ units are then g kg$^{-1}$ m s$^{-1}$. However, in giving the final output, we convert L m$^{-2}$s$^{-1}$ (equivalent to mm) once grams are transformed to liters and dry air density (kg m$^{-3}$) is included as:

$$W_h = r_l \, \rho_a \, u_x \, \eta \qquad (6)$$

Finally, $W_h$ is integrated over a period as:

$$\overline{W}^{\Delta t}{}_h = \int_{t0}^{t1} W_h\, \partial t \;,\qquad (7)$$

where $t_0$ and $t_1$ correspond to the initial and ending times. The model has three main assumptions described as follows: (1) $F_{in} > F_{out}$; (2) since the model aims to reproduce advective fog collection, it is assumed that condensation only occurs in the atmosphere under the conditions $r_l = r_v - r_s$; (3) the mixing ratio ($r_v$) being two orders of magnitude higher than $r_l$, is nearly conserved.

In Equation (6), $r_l$ and $\eta$ depend on location and condensation processes. Regarding location, $r_l$ varies in height (the vertical dimension of the model) and depends on the conditions of the marine Sc cloud over the ocean and its interaction with the topography. To estimate this variable using routine data, we assume that water vapor condenses once it reaches the thermodynamic conditions to reach saturation, This assumption implies that we do not take microphysical properties such droplet size, nucleation or droplet concentration into the calculations. The second term, $\eta$ groups cloud microphysics, the collector design, and its material properties. To delve into the detailed calculation of $r_l$ and $\eta$, we break down the analysis of Equation (6) into two parts: the thermodynamic and water potential modules (section 2.1 and 2.2). In addition, we introduce a third module for representing the model's horizontal spatial variability of $W_h$ trough spatial interpolation creating a fog harvesting potential map."

• Introduction is given in a large parag that doesn't focus on fog physics/dynamics etc at all. There are several works on marine fog (Gultepe et al 2021 BLM; 2019 Marine fog review; Fernando et al 2021) that are not mentioned. Characteristics of LWC, Nd, and DSD are provided in these works.

In the introduction we keep a balance between the essential physics of the transition marine stratocumulus to land fog over coastal mountains, and the quantification of the water yield. Following this argument, the manuscript describes the marine stratocumulus formation between lines 43-57 as the main mechanism of fog formation. To gain in clarity we have revised the introduction. In short, we have divided the instruction in three parts: 1) problem statement, 2) fog-cloud dynamics and 3) fog collection. In the second part we have focused on fog with similar characteristics as the one studied. Specially, we have introduced the following changes (lines 50 to 55), to reinforce the physics and dynamics of fog, including microphysical processes and the suggested references provided by the referee.

"Here, one of the main physical involved in stratocumulus formation is the microphysical properties of cloud droplets, which are linked to cloud optical properties that have important climate effects (Wood, 2012). In the South East Pacific, cloud droplet sizes of 5 to 15 μm are often found, whose concentration is ≤50 cm$^{-3}$ increasing to 200 cm$^{-3}$ along coastal areas of Chile (Painemal et al., 2011). The droplet size and concentration determines the liquid water content (Gultepe et al., 2021), which essentially is the amount of water that can be harvested on land once Sc becomes fog."

• Fog device for LWC is being used since 2006 (Gultepe et al BAMS and others).

We have introduced the fog measurement devices and their respective reference in line 135.

• Where is the importance of Nd in the model? Without Nd, how do you get accurate LWC? What is physically missing here? What is the role of Nd in LWC?

As mentioned in our general comment, the model assumes an "all or nothing" formation of liquid mixing ratio. The satisfactory evaluation we show in sections 2.1 and 2.1.4 with the available observations reinforces that this simplistic assumption gives physical consistent estimations in magnitude, height-dependence, and evolution. However, as the referee mentioned, further

improvements to the model will aim to increase the complexity of the microphysical mechanism, providing that we have enough observational evidence.

We introduced several changes in Section 4 (**between lines 446 and 451**) by including:

"Firstly, one of the most important variables in the model is the adiabatic liquid water mixing ratio ($r_l$), which is estimated assuming water vapor is condensed because it reaches saturation. Despite our simplistic approach and reliable results, we know that further model improvements must be made by including essential microphysical processes. Such processes are mean volume diameter, effective size, droplet concentration, and effective droplet size (Gultepe, et al 2021). To account for these processes, comprehensive observations must be performed to get a complete budget equation allowing us to have more realistic modeling."

Moreover, conclusions have been restructured including now the just mentioned limitations (See comments below).

• See for budget equations given for cirrus clouds in PAAG 1990s Gultepe et al where steady state assumption is assumed to solve hor and ver advections. In addition, what role turbulence, radiation, and vertical advection play role? What happened to these terms?

We would like to answer the referee with a question: Can this budget terms be explicitly and accurately calculated in an area characterized by a strong ocean-land contrast and a very strong slope (1000 meters in the first 5 km)? We agree with the referee that our research should indeed aim to get a complete budget as s/he is mentioning. However, in absence of complete and comprehensive observations in space and in time, or a more explicit simulations using large-eddy simulation accounting with strong height differences, we have opted for a simple conceptual model to attempt to understand the main physics and reproduce the available observations.

• Fig. 7 suggests that there are huge diffs between observed and model simulations per month, how can be annual values get closer so much? Something is wrong.

Regarding Figure 7, we have revised the observations used for the model comparison, as suggested by the first reviewer. Our thorough revision revealed that some observations of sites B (arid) and C (semi-arid), were seriously affected by broken meshes and pipe's obstructions. Therefore, we decided to exclude such months in order to make a fair comparison with modeling results. The new Figure 7, which includes results obtained from the model reformulation and quality-filtered observations, is shown below.

[Figure]

Here, we observe that monthly and annual daily averages modeled collected water in the three sites are in 80% agreement with observations. In addition, the model is able to reproduce the seasonal cycle in magnitude and variability.

• How do you use satellite obs is not clear, how do you get fog LWC/LWP or coverage, no method is given properly.

As explained in section 2.3, the vertical variability of modeled water harvesting ($W_h$) is spatially extrapolated using as a base map the fog and low cloud (FLC) frequency image obtained through the GOES satellite (Espinoza et al., 2024) and a digital elevation model (DEM). We are not obtaining the liquid water content or liquid water path through the satellite, but using GOES+DEM as a geographical framework for extrapolating the vertical fog harvesting potential ($W_h$). This simple spatial extrapolation aims to know the horizontal variability that vertical fog harvesting potential might have.

We separate this extrapolation into four steps:

1. Reclassifying DEM grid cells whose height is between the cloud base and cloud top height determined by the model.

2. Reclassifying DEM grid cells with orientation based on main (mean ± std) wind direction observed at the $z_2$ station.

3. Calculating the FLC frequency using GOES satellite and intersect grid cells obtained in steps 1, 2, and 3.

4. As both the intersect grid cells from the last step and the vertical variability of $W_h$ have a vertical domain (height), we replace the altitude values of the grid cells with $W_h$ values, resulting in a spatial variability of $W_h$.

To gain clarification in the manuscript, we have introduced several changes in Section 2.3:

"In addition to the thermodynamic module, we propose a spatial module for extrapolating the vertical variability of $W_h$ into a horizontal spatial domain. To do it, we integrate the vertical domain (z) of $W_h$ to an area of optimal fog harvesting potential obtained from a combination of a digital elevation model (DEM) and GOES satellite images. We outline four steps to achieve this spatial variability.

The first step involves reclassifying the DEM grid cells based on the cloud layer height and removing all grid cells below the CB and above the CT elevation. This reclassification ensures that only the elevation range where the Sc cloud could potentially impact the topography is considered. In the second step, we create an aspect image (slope orientation) with the DEM and reclassify the pixels based on the angle range of the main wind direction (mean ± std) when fog is collected (obtained from observations at the $z_2$ station). The third step involves calculating the fog and low cloud (FLC) frequency using data from the GOES satellite (del Rio et al., 2021; Espinoza et al., 2024). This algorithm continuously calculates the presence and absence of FCL in every GOES grid cell. The third step serves as a geographical framework, delineating the area where fog-cloud interacts with topography. The spatial intersection of the three steps generates optimal areas for fog collection, physically representing the locations where the Sc cloud and its harvesting potential intersect the surface. It is important to note that the values of grid cells in these optimal areas for fog collection represent elevations (m ASL) in areas with high FLC frequency. The final step involves replacing the elevation grid cell values of the optimal fog collection areas with the vertical distribution of potential fog harvesting ($W_h$). As $W_h$ values are associated with a vertical domain (z), each $W_h$ value can be mapped onto the resulting grid of optimal fog collection areas. The result of this last step yields a spatial distribution of potential fog harvesting."

• Apply these results for climate is very much simplification, this should be taken out.

We agree with reviewer and we have removed the following sentence in line 475 to 476 (from the original manuscript) from the conclusions:

"Moreover, by using future climate data projections, it becomes feasible to assess how these water resources might respond to climate change. "

And line 477:

"and the study of the climatological evolution of cloud water, among others"

• Conclusions; needs to be collected for a few bullets and explained based on the text, not clearly explained properly.

We agree with the reviewer and following his/her advice we have restructure the conclusion in relevant bullet points focus on:

- Model reliability to reproduce in time and space fog harvesting despite its simple approach and limited data used.

- Model limitations, challenges, and further improvements.

- Potential uses of this model for water planning.

These major changes have been introduced in the conclusion as follow:

"We propose, formulate, and evaluate an observational-driven model, named AMARU, for estimating advective fog water potential harvesting in (semi-)arid regions. This model uses standard and routine meteorological observations to estimate where, when, and how much water can be potentially

harvested from fog clouds. The proposed model employs a thermodynamic approach to estimate fog's adiabatic liquid water mixing ratio, incorporating key physical processes associated with the interaction between the stratocumulus cloud and topography. This approach yields vertical profiles of liquid water mixing ratio, from which fog frequency, cloud base, and top can be derived. In addition, by integrating the estimations of liquid water mixing ratio with climatological records of fog harvesting observations, we derive an empirical collector efficiency coefficient to estimate vertical profiles of potential fog harvesting. Finally, combining vertical profiles of fog harvesting potential with satellite products, we introduce a methodology for spatially extrapolating these results, thereby generating fog harvesting potential maps.

Below, we outline the main conclusions of our research.

- Despite the simple approach, this model correctly reproduces essential physical components involved in fog harvesting. Our evaluation with available observations show that model results reproduce: fog frequency (R: 0.95; RMSE: 6%), cloud base and top height (errors <50 m), liquid water content (errors ~0.2 g m$^{-3}$), and fog collector efficiency (errors ~5%). Overall, fog harvesting observation are satisfactorily reproduce by the model with mean errors of 10% (<1 L m$^{-2}$).

- The simple approach takes advantage of using routine meteorological data, which is widely available worldwide in areas characterized by land-ocean contrast and complex topography.

- However, the model presents several limitations, whose improvement will depend on comprehensive observations and further research. Between the limitations, microphysics observations of cloud droplet size, concentration, and actual water content must be incorporated to improve the model. Moreover, further research must be done on the empirical coefficient, which is constant in the model. However, our observations suggest a variability which depends mainly on wind speed, but also in the materials. Finally, future research should incorporate accurate vertical profiles of temperature, mixing ratio, and wind speed to corroborate our vertical modeling assumptions.

- Our model offers a versatile approach with multiple applications in massive fog harvesting planning and ecosystem delimitation for conservation purposes, among others. Since fog is a global meteorological phenomenon, this model holds potential for applicability in many coastal (semi-)arid regions, addressing data deficiencies in regions where fog harvesting represents a viable water source.

Finally, we expect this research to yield significant social benefits by providing decision-makers with valuable insights into new water sources, thus aiding in the mitigation of climate change impacts."

• Appendix also has severe issues without providing assumptions. Overall, I cant accept this paper as scientifically meaningful and it needs lots of work to be published.

As mentioned in this point-to-point answer we have attempted to clarify the derivation of the model and explain assumptions (resulting in removing the Appendix). Here, we would like to stress the original aspects of our research: a conceptual model that is able to provide an interpretation of the transition stratocumulus to fog in a very complex topographic area. The possibility to use this model on the interpretation of diurnal variability (height), but also seasonal to yearly variations. Finally, the possibility of using this model as a predictor using routine and standard meteorological variables, either observations or model results.

---

## Referee Report (RR1)

Observation-driven model for calculating water harvesting potential from advective fog in (semi-)arid coastal regions. By Felipe Lobos-Roco, Jordi Vilà-Guerau de Arellano, and Camilo del Río.

General comments.

Most of my concerns with the initial version have been addressed and the AMARU model appears to be useful for practical estimation of water harvesting. There are some places in the revised manuscript where minor changes could be made to improve clarity and some suggestions are given below. One concern would be Equation 12. Once cloud or fog has formed I would expect 100% relative humidity and, with $r_v = r_s$, Eq (12) would give zero liquid water mixing ratio. I think we are to assume that $r_v$ is the surface level ($z_1$) mixing ratio and represents the mixing ratio of the air before the stratus cloud had formed.

Detailed comments

line 8              "is $\leq 50$ cm$^{-3}$ 50 cm$^{-3}$ " repetition.

line 56             "how well-mixed ($<3.1$ x $10^{-3}$ K$^{-1}$) the MBL.."  Make it clear that this is $\partial\theta/\partial z$, and introduce "potential temperature".

line 59             "As the latter increases, the liquid water content progressively..."  The LCL or just height?

line 90             If $F_{in}$ is a flux it should be per unit area"

line 97             Why add in std?

line 119            "through"

line 124            "devices"

line 140            maybe ".. the potential temperature gradient ($\partial\theta/\partial z$)" ...

line 142/3          Is q the (water vapor) mixing ratio or specific humidity? Virtually the same but best to be consistent. Also well mixed potential temperature.

Figure 3            Units for  $\partial\theta/\partial z$ and $\partial q/\partial z$ thresholds.

line 150            Case 4 is a dewpoint depression of 1.15K, not 1.5 K?

line 159/160        If there is no fog $\partial q/\partial z$ could have any value and once fog is formed q = qs (saturation mixing ratio), but T dependent and not well mixed.  I am nor sure what to read into +not contingent"

line 179            I suspect a typographical error in Equation (8), One of the "1-" expressioins should probably be removed!

line 189/190        the "uplifting" in Fig 2C will depend in part on the topography - airflow over the mountain or through gaps?  Is the 3D terrain structure taken into account?

p10,11              CB estimates seem better than CT. CT will depend on many factors, such as the initial humidity profile when the stratus clouds were forming.

line 242            Need more explanation of the basis for Equation(12).
* * *
My apologies for taking so long to get to this - the past few months have been busy, and this was all I had time for.  Still Oct 7 EDT, just!

---

## Author Response (AR2)

**Reply to reviewers - Second round**

**Review of HESS 2024-110**

Observation-driven model for calculating water harvesting potential from advective fog in (semi-)arid coastal regions.

By Felipe Lobos-Roco, Jordi Vilà-Guerau de Arellano, and Camilo del Río.

**Reply to Editor**

Editor decision HESS-2024-110

Dear authors, two referees have assessed the revisions you made to the manuscript, one of them being mostly satisfied except a couple of remaining remarks, the other not entirely. When reading the revised manuscript myself I ran into a couple of issues that made it hard to understand the reasoning. I ask you to address the comments and suggestions made by reviewer #1 and address additional comments as follows (line numbers refer to manuscript version with tracked changes):

Dear editor, we appreciate very much that you took the time in reading and reviewing the paper. We have read the comments of reviewer #1 and your own comments and we have carefully addressed all of them, and we have modified the paper accordingly. Below, in blue font, you will find our responses, including the line numbers where changes were made in the revised version of the manuscript.

– Line 51: "one of the main physical involved in.." : noun missing after physical

We have modified the sentence as follows: "Here, one of the main physical **processes** involved in stratocumulus formation is the microphysical properties of cloud droplets,".

– Line 53: <50cm$^{-3}$ 50 cm$^{-3}$ …Is there a typo here?

Agree, the misspell have been corrected.

– Line 56-57: please explain why the MBL has to be well mixed for advective fog formation? You provide a very strict limit for well-mixed versus stratification (<3.1 $10^{-3}$ K/m) – how accurate and how generally valid is this limit? Please clarify.

The sentence on lines 56-57 indicates that stratocumulus formed and maintained over the ocean, move (advect) over land and lead to fog formation. Since conserved variables are well-mixed below the stratocumulus deck, we assume the same condition over land. Regarding the thresholds mentioned ($\partial\theta/\partial z$ and $\partial q/\partial z$), Lobos-Roco et al. (2018) studied these in the Atacama region based on observation of fog collection and meteorological data on a topographic transect. In the study, the authors define two MBL layer regimes related to fog formation and dissipation when MBL is either well-mixed or stratified. These thresholds were calculated during nine advective fog events in the Atacama region during all seasons in 2015 and validated using fog collection observations.

– line 60-61: you state that liquid water content peaks at the cloud top at ~0.7g kg$^{-1}$ Same question here: please clarify how generally valid is this statement

0.7 g kg$^{-1}$ is the maximum mean liquid water content observed on a vertical profile over marine stratocumulus in several studies. For example, Duynkerke et al. (1995) in the north Atlantic and recently Schween et al. (2022) in the coastal Atacama region. To clarify the sentence we have modified it as follows:

"From the lifting condensation level to up, the measured liquid water content progressively rises. Based on observations in the same region, we take 0.7 g kg$^{-1}$ at cloud top as the maximum value (Schween et al., 2022)"

– In section 2 (lines 85-90): please explicitly state the aim of the model here.

We have modified the sentences in line 85-90 as follows:

"The AMARU reproduces the fog that can be potentially harvested using standard fog collectors, estimating the liquid water content of the air. A particular aspect of AMARU is to apply the available routine meteorological observations to obtain this liquid water content. The model is based on the evolution of time and the height of marine stratocumulus adiabatic liquid water content moving towards land characterized by complex topography."

– For the equations: make sure to explain all symbols used in the equations and clarify their units

Thanks, we have checked and made small changes to clarify.

– The term "fog occurrence" is used a lot without explaining the reference time period for which the frequency is calculated (frequency is typically defined as nr of occurrence per time period). Please provide a clear definition of fog occurrence

We understand the Editor's concern to make it clearer, we have added the following sentence at line 140-142:

"We define fog frequency as the number of counts when fog is present over a timestep (in an hour), expressed in percentages. For example, 50%'s fog frequency means that fog occurred during 30 min over an hour. "

– Figure 3: clarify what data (and model outputs) are behind the statistics shown in the Taylor diagram, incl the number of data points used. In the figure caption clarify which "line" you mean in "Note that … placing behind the line and …."

Agree, we have modified the caption of Figure 3 as follows:

"Figure 3. (a) Taylor diagram comparing the proposed criteria and thresholds for estimating fog frequency (%). The diagram deploys correlation (r), standard deviation (in FF units, %), and root mean square error (RMSE in FF units, % %) between the criteria-thresholds and observations. The number of data points used is 8760, which corresponds to hourly data over a year. (b) Comparison of the annual diurnal cycle of fog occurrence between observations (SFC) in blue and the best-performing criteria in black. Every black/blue mark represents the presence (100% of frequency) at every hour during 2018. Note that numbers 11 and 12 have a slightly negative correlation, placed behind (left) to the Taylor diagram y-axis."

– Section 2.1.1 Fog cover fraction frequency. Consider using a different term. What is a "fraction frequency.."?

The term "fog cover fraction frequency" was thought to be like cloud cover fraction. However, we agree it can induce confusion, so we have modified it simply as "fog frequency".

– In 2.1.1. percentages are used to for occurrence (In line 156: fog occurrence (in %) – why not frequency as before? Especially because you % a lot in section 3 to discuss disagreements between observations and model. It can be confusing..

We agree, we have defined fog frequency to avoid this confusion.

– Equations 9 and 11: are you sure these constants are accurate enough to use so many digits? Also, make sure to explain how these constants were determined.

The constants result from the linear regression model obtained from the relation between the observed cloud base and top during the GOFOS experiment. We decided to use the full decimals to be more accurate. To be more explicit, we have modified the following sentence in line 229-230:

"Equation 9 shows a linear regression model in which CT [m] solely depends on CB [m], where constants are determined from the relation between observed CB and CT during the GOFOS experiment.".

– Section 3, line 356: "disagreements with observations by 1% to 20%. The units in the figure are not %, so these percentages refer to what, exactly..?
–

In section 3 line 356, the percentages between 1% to 20% refers to the differences of annual daily rates of fog collection between model and observations. This is explained in the lines after, where, for example, observations in site a is 5.5 L $m^{-2}$ and model 5 L $m^{-2}$, then the model disagrees by 10%. To avoid confusion with frequency and efficiency, which are also in %, we decided to leave the comparison here in L $m^{-2}$.

– line 361, 362: errors around 0.39 $lm^{-2}d^{-1}$(20%), (…) errors of 2 $lm^{-2}d^{-1}$ (18%). Where exactly do these values come from? What do the %-s mean, % of what?

To avoid confusion, we will remove the comparison in % and only leave the ones in L $m^{-2}$. These numbers refer to the differences between monthly averaged fog collection daily rates (L $m^{-2}d^{-1}$), shown in Figure 7. The 0.39 L $m^{-2}d^{-1}$ is the mean difference between model and observations for summer months (January and March), whose difference is about 20%. .

– Line 364: observed rates 4 to 9 (..) – I can't relate this range to the figure, for site A? How did you find the range that you're reporting here?

These numbers 4 to 9 and 6 to 10 L $m^{-2}d^{-1}$ refer to the error bar length (black vertical line over the plot) during spring (September to November).

By reading to the last three comments, we have introduced several changes in section 3.1 to explain the comparison between observations and modeling clearer and more explicit.

General comment: make sure to check spelling and grammar throughout the document. There are many instances where plural is used instead of singular and vice versa, in some sections past and present tense are mixed.

Agree. To improve the readability, we have re-checked the entire manuscript looking for spelling errors with a native English language editor.

**Reply to Reviewer #1**

General comments.

Most of my concerns with the initial version have been addressed and the AMARU model appears to be useful for practical estimation of water harvesting. There are some places in the revised manuscript where minor changes could be made to improve clarity and some suggestions are given below.

We are grateful to have addressed most of the reviewer's initial concerns and thank him for his valuable contribution to improve our manuscript through the alternative analysis. Regarding the new minor comments, we have accepted all the reviewers' comments, corrections, and suggestions, resulting in the corresponding changes in the manuscript. Below, in blue font, you will find our responses, including the line numbers where changes were made in the revised version of the manuscript.

Detailed comments

— line 8 "is ≤50 cm −350 cm −3 " repetition.

Agree, the misspell have been corrected.

— line 56 "how well-mixed (<3.1 x 10 -3 K -1 ) the MBL.." Make it clear that this is $\partial\theta/\partial z$, and introduce "potential temperature".

Agree, we have modified the sentence as follows:

"Formation and maintenance depend on how well-mixed the MBL is in terms of potential temperature ($\partial\theta/\partial z < 3.1$ x $10^{-3}$ K m$^{-1}$) the MBL is, while the dissipation is influenced by its stratification ($\partial\theta/\partial z > 3.1$ x $10^{-3}$ K m$^{-1}$) (Lobos-Roco et al., 2018)."

— line 59 "As the latter increases, the liquid water content progressively..." The LCL or just height?

The way the phrase is written results in confusion, we decided to modify as follows:

"From the lifting condensation level to up, the measured liquid water content progressively rises. Based on observations in the same region, we take 0.7 g kg$^{-1}$ at cloud top as the maximum value (Schween et al., 2022)"

— line 90 If F in is a flux it should be per unit area"

The unit of the flux is g kg$^{-1}$ m s$^{-1}$, which after multiplied by air density it ends as L m$^{-2}$ s$^{-1}$, as is explained in line 103.

— line 97 Why add in std?

We understand the reviewer's concern. Mathematically, we should use the perpendicular mean wind ($\overline{u_x}$). However, in reality, fog can also be collected with a tilted wind respect to the perpendicular. To include this effect, we add the standard deviation to amplify the range of fog influx.

— line 119 "through"

Agree, corrected.

- line 124 "devices"

Agree, corrected

- line 140 maybe ".. the potential temperature gradient (∂θ/∂z)" …

Agree, corrected

- line 142/3 Is q the (water vapor) mixing ratio or specific humidity? Virtually the same but best to be consistent. Also well mixed potential temperature.

Agree, corrected

- Figure 3 Units for ∂θ/∂z and ∂q/∂z thresholds.

Agree, The unit thresholds have been included

[Figure]

- line 150 Case 4 is a dewpoint depression of 1.15K, not 1.5 K?

Agree, corrected

- line 159/160 If there is no fog ∂q/∂z could have any value and once fog is formed q = qs (saturation mixing ratio), but T dependent and not well mixed. I am nor sure what to read into +not contingent"

We agree that this sentence leads confusion. We decided to delete it to improve the paragraph readability.

- line 179 I suspect a typographical error in Equation (8), One of the "1-" expressioins should probably be removed!

Agree, the equation has been corrected as it is in Lobos-Roco et al., (2018)

- line 189/190 the "uplifting" in Fig 2C will depend in part on the topography - airflow over the mountain or through gaps? Is the 3D terrain structure taken into account?

The uplifting shown in Figure 2C depends on the topography where the $z_2$ station is located. In the manuscript, we test the model over three sites where the $z_2$ station is located over the mountain and

not through gaps (valleys). Then, the uplifting results from the combination of potential temperature between $z_1$ and $z_2$. Here, if $z_2$ is warmer than $z_1$, then the combined potential temperature of MBL ($z_1$ and $z_2$) will also be warmer, and consequently, LCL will be reached at a higher height. For calculating the vertical profiles of $r_l$ or $W_h$, the model do not take into account 3D topography. However, as explained in section 2.3, we combine vertical profiles of $r_l$ or $W_h$ with topographic 3D model and the GOES fog and low cloud algorithm to approach to represent fog collection in the space.

— p10,11 CB estimates seem better than CT. CT will depend on many factors, such as the initial humidity profile when the stratus clouds were forming.

We agree with the reviewer that CT will depend on many factors. However, for simplicity, in this model, we opted to estimate the cloud top based on the relation between observed CB and CT (equation 9), which we have improved by including two factors: fog frequency (equation 10) and potential temperature gradient (equation 11). We are aware that CT calculation needs to be improved in further research with more accurate CT measurements, which is discussed in section 4 (model limitations and challenges).

— One concern would be Equation 12. Once cloud or fog has formed I would expect 100% relative humidity and, with $r_v = r_s$, Eq (12) would give zero liquid water mixing ratio. I think we are to assume that $r_v$ is the surface level ($z_1$) mixing ratio and represents the mixing ratio of the air before the stratus cloud had formed.

— line 242 Need more explanation of the basis for Equation(12).

Equation 12 is only applied above the LCL. LCL is defined by RH=100 ($r_v$-$r_s$=0). As a first approximation ($r_v$ is not a conserved variable, but very close to a conserved variable specific humidity), $r_v$ is constant with height. Between LCL and cloud top, any excess of $r_v$ with respect the change of $r_s$ (T) results in $r_l$.

We have introduced the following changes in line 253-256 as follows:

"where $r_v$ is the mixing ratio between grams of mass water vapor by kilogram of dry air, $r_s$ is the saturated mixing ratio, and z represents the vertical level between CB and CT (Fig. 2a). Here, since $r_v$ is very close to be a conserved variable ($r_v$~$q_v$), it is assumed as constant over the cloud layer. Therefore, any excess of $r_v$ with respect the change of $r_s$ (T) will result in $r_l$."